# Shapes and Rise Velocities of Single Bubbles in a Confined Annular Channel: Experiments and Numerical Simulations



Andrea Cioncolini [1,*] and Mirco Magnini [2,*]

1   Department of Mechanical, Aerospace and Civil Engineering, University of Manchester, Oxford Road, Manchester M13 9PL, UK
2   Department of Mechanical, Materials and Manufacturing Engineering, University of Nottingham, University Park, Nottingham NG7 2RD, UK
*   Correspondence: andrea.cioncolini@manchester.ac.uk (A.C.); mirco.magnini@nottingham.ac.uk (M.M.)

**Abstract:** Shapes and rise velocities of single air bubbles rising through stagnant water confined inside an annular channel were investigated by means of experiments and numerical simulations. Fast video imaging and image processing were used for the experiments, whilst the numerical simulations were carried out using the volume of fluid method and the open-source package OpenFOAM. The confinement of the annular channel did not affect the qualitative behavior of the bubbles, which exhibited a wobbling rise dynamic similar to that observed in bubbles rising through unconfined liquids. The effect of the confinement on the shape and rise velocity was evident; the bubbles were less deformed and rose slower in comparison with bubbles rising through unconfined liquids. The present data and numerical simulations, as well as the data collected from the literature for use here, indicate that the size, shape, and rise velocity of single bubbles are closely linked together, and prediction methods that fail to recognize this perform poorly. This study and the limited evidence documented in the literature indicate that the confinement effects observed in non-circular channels of complex shape are more complicated than those observed with circular tubes, and much less well understood.

**Keywords:** bubble; shape; rise velocity; confinement; annular channel; experiment; simulation





## 1. Introduction

The dynamics of single bubbles rising through stagnant liquids is of pivotal importance for the fundamental understanding of two-phase bubbly flows, which are relevant in a number of applications including gas–liquid reactors, bubbly columns, nuclear reactors, heat exchangers, and environmental flows. Gas–liquid contacting, which is normally achieved by bubbling a gas into a liquid, is in fact one of the most common operations in the process industry in applications such as absorption, distillation, and froth flotation.

Despite the apparent simplicity, the dynamics of single bubbles rising through stagnant liquids is in reality a rather involved free-boundary problem controlled by the interplay among inertia, buoyancy, viscous, and surface tension forces. When bubbles rise in bounded liquids their dynamics are also affected by the walls of the container. Available experimental studies of single bubbles rising through stagnant liquids were performed in containers of finite dimensions, typically vertical tanks of either circular or square cross-sections. Strictly speaking, therefore, wall effects were always present to a greater or lesser extent. However, it is currently accepted that, if the dimensions of the horizontal cross-section of the container are much larger than the size of the bubble (indicatively, 10–20 times or more), then wall effects on the bubble dynamics are small or absent. The bubble can therefore be considered unconfined, and the observed bubble dynamics can be regarded as representative of the free bubble rise through a stagnant unconfined liquid. This is the case for the majority of the experimental studies documented to date [1–3].

Wall effects were investigated rather extensively for bubbles rising through circular tubes [4–6] and through narrow rectangular channels [7–13]. For circular tubes, wall effects typically cause the elongation of the bubbles in the vertical direction and the alteration of the wake structure behind the rising bubble, resulting in milder bubble deformation and a reduced rise velocity in comparison with the unconfined case. For narrow rectangular channels, bubbles are typically bigger than the channel gap, so that their dynamic is largely controlled by wall effects.

In contrast, the dynamics of single bubbles rising through confined channels of more complex shape, representative or informative of rod bundles or bubbly columns with internals, were not extensively investigated; the only studies documented in the literature appear to be those by Venkateswararao et al. [14] and by Tomiyama et al. [15]. In particular, Venkateswararao et al. [14] measured the rise velocity of air bubbles rising in stagnant water (tap filtered) at ambient conditions through a rod-bundle column comprising sixteen rods of 12.7 mm diameter, arranged in a square lattice, with pitch of 17.5 mm, and installed inside a 88.9 mm diameter circular pipe (corresponding to a subchannel hydraulic diameter of 18.0 mm). Partial rods were placed along the inner surface of the circular pipe to minimize end effects and make the cross-section of the test piece representative of a larger rod bundle. For bubble sizes in the range of 2–8 mm, the measured rise velocities increased with the bubble diameter: from about 0.21–0.22 m/s for 2–3 mm diameter bubbles to about 0.3 m/s for 7–8 mm diameter bubbles. On the other hand, Tomiyama et al. [15] measured the shapes and rise velocities of air bubbles rising in stagnant water (distilled) at ambient conditions through an inner subchannel comprising four rods of 12.0 mm diameter, arranged in a square lattice, with a pitch of 15.2 mm (corresponding to a subchannel hydraulic diameter of 12.5 mm), and installed inside a square pipe. The rods were in contact with the inner surface of the square pipe. For bubble size in the range of 3–6 mm, the measured rise velocities decreased with bubble diameter: from about 0.21–0.22 m/s for 3 mm diameter bubbles to about 0.17–0.18 m/s for 6 mm diameter bubbles.

Therefore, as can be noted, the documented studies of single bubbles rising through confined channels of complex shape are not only a few in number, they also yield conflicting results regarding the bubble rise velocity, which increases with bubble diameter according to the measurements of Venkateswararao et al. [14], whilst it decreases with bubble diameter according to the measurements of Tomiyama et al. [15]. This clearly indicates that more investigations are needed on single bubble rise in non-circular channels of complex shape, thereby creating the motivation for the present work. In this study, we investigated by means of experiments and numerical simulations the shapes and rise velocities of single air bubbles rising through stagnant water inside an annular channel, a geometry not previously considered in single bubble studies. In particular, we used the numerical simulations to help interpret the data and widen the scope of the experimental observations.

## 2. Experiments

### 2.1. Experimental Setup

A schematic representation of the experimental setup is provided in Figure 1a,b. The test piece is a vertical annular channel with a hydraulic diameter of 16.9 ± 0.5 mm, comprising a square cross-section transparent pipe (high-temperature-rated clear acrylic) with nominal inner size of 25 mm (measured actual size of 25.3 ± 0.3 mm), which accommodates an internal stainless-steel rod of 10 mm nominal diameter (measured actual diameter of 9.99 ± 0.01 mm). The present experimental setup was specifically designed to carry out three different types of experimental investigations:

1. Single air bubble dynamics in confined stagnant water;
2. Multiple air bubble dynamics in confined stagnant water with annular channel operating as a small bubble column running in batch mode;
3. Pool boiling in water where bubbling is sustained via Joule heating of the internal rod.

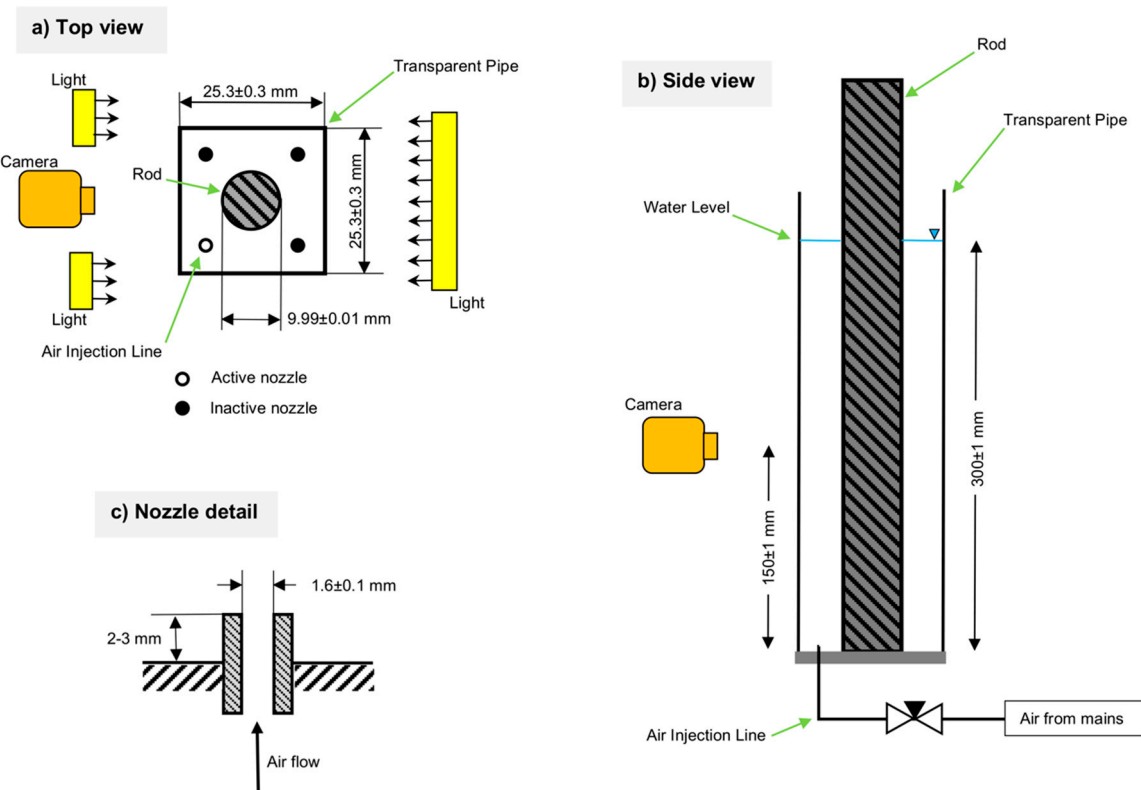

**Figure 1.** Schematic representation of the experimental setup (vertical dimension not to scale): (**a**) top view (as explained in Section 2.1, out of the four injection lines present only one was used whilst three were inactive), (**b**) side view (only the active injection line is shown), (**c**) detail of the nozzle tip used to generate the bubbles.

Only the results of the single bubble dynamics experiment from Point 1 are documented herein; the results of the other experiments will be communicated separately (the experiments on multiple air bubble dynamics are scheduled for the first and second quarters of 2022, whilst the experiments on pool boiling are scheduled for the last quarter of 2022). Nonetheless, it is important to clarify beforehand the wider scope of the present experimental setup, whose design is a compromise among the different requirements of the experimental investigations listed above. Clearly, this has informed the experimental procedure adopted here.

Regarding the single bubble dynamics experiments considered here, the testing fluids were micro-filtered and high-purity water (conductivity of 0.005 µS/cm) and filtered air from the mains. The water depth level in the annular channel was of $300 \pm 1$ mm, and was not changed during the tests. Bubbles were generated by blowing air through a capillary pipe (diameter of $1.6 \pm 0.1$ mm) protruding at the bottom of the annular channel, as schematically illustrated in Figure 1c. The length of the capillary pipe was of 250 mm (corresponding to a length-to-diameter ratio of 156), which is large enough to ensure that the pressure fluctuations due to the formation of the bubbles have a negligible effect on the air flow rate, which can then be taken as constant during the tests. The air flow rate through the capillary pipe was adjusted with a needle valve to yield a bubbling frequency of 0.45 Hz (corresponding to one bubble generated every 2.2 s), which was not varied during the experiments. This yielded bubble diameters of about 4.3 mm. The residence time of the bubbles in the column is below 2 s, thereby ensuring that, when a bubble is generated at the nozzle, the previously generated bubble has already reached the top of the channel. As confirmed by the CFD simulations discussed later, this provides enough time for the liquid to recover between successive bubble passages, so that each bubble rises through a virtually still body of water. The present set-point with a bubbling frequency of 0.45 Hz, therefore, ensures continuous bubbling at a steady rate with a delay between

successive bubbles sufficient to allow liquid recovery and, correspondingly, treat each bubble as a single bubble rising through a stagnant liquid.

In the provision for the experiments on multiple bubble dynamics (Point 2 above), the annular channel in Figure 1 was equipped with four independent and identical air injection lines. However, only one air injection line was used for the single bubble experiments reported herein, whilst the other three air injection lines were inactive: the four nozzles, one active and three inactive, are shown in Figure 1a.

The present bubble generation method differs from that commonly employed in single bubble studies, where bubbles are generated with syringe pumps. These latter are clearly more flexible, and allow a range of bubble sizes to be generated from a nozzle of fixed diameter. In contrast, in the present setup, the bubble size does not vary once the air flow rate is fixed. To overcome this limitation, we used CFD simulations (discussed later) to generalize the experimental observations to bubble diameters in the range of 3–6 mm. The present bubble generation method was chosen because it was more representative of bubble columns which, as explained before, were within the broader scope of the experimental facility. One disadvantage of bubble generation with syringe pumps is that the initial deformation of the bubbles is variable: the larger the bubble generated from a nozzle of given diameter, the larger the initial deformation of the bubble [16,17]. Particularly with air–water systems and for bubbles in the range of about 1–10 mm in diameter, where surface tension plays a dominant role on the bubble dynamics, the initial deformation of the bubble can have a profound influence on the subsequent bubble rise: the bigger the initial deformation, the bigger the subsequent rise velocity [16–20]. The consequence is that bubbles of comparable size generated with syringe pumps from nozzles of different diameters can have different initial deformations, and, subsequently, exhibit different rise velocities. This is in fact one of the reasons behind the large scatter in the documented results for the rise velocity of air bubbles in water [1], the other reason being the presence of contaminants in the water. With the bubble generation method used here, the initial deformation of the bubbles is not variable and the scatter in rise velocity is not observed, as described later.

Measurements were carried out at ambient pressure and room temperature ($295 \pm 2$ K). The relevant properties (liquid and gas densities $\rho_l$ and $\rho_g$, liquid and gas viscosities $\mu_l$ and $\mu_g$, and surface tension $\sigma$) of the testing fluids were calculated with NIST-REFPROP [21], and their values are provided in Table 1. As noted previously, during the tests, the water depth level in the annular channel was $300 \pm 1$ mm. The variation of the air density due to the hydrostatic pressure variation along the channel is within a few percent, and thus can be neglected.

**Table 1.** Relevant properties of water-air at 295 K.

| $\rho_l$ (kg/m³) | $\rho_g$ (kg/m³) | $\mu_l$ (µPa s) | $\mu_g$ (µPa s) | $\sigma$ (mN/m) |
|---|---|---|---|---|
| 998 | 1.18 | 958 | 18.3 | 72.4 |

Despite its simplicity, justified by the scope of the present study, which is more of fundamental rather than applied character, the present experimental setup can be informative of more complex configurations of direct industrial relevance, such as rod bundles and bubble columns with internals.

*2.2. Measurement Methodology*

Rising bubbles were recorded using a digital camera (Bonito CL-400 equipped with a Navitar 25 mm Platinum F8 lens—image resolution: $864 \times 1168$ pixels; recording frequency: 386 fps) providing a spatial resolution of 35.4 µm/pixel (corresponding to 28.2 pixel/mm), appropriate for resolving the present bubbles, which are a few mm in size. As shown in Figure 1b, the digital camera was positioned midway through the annular channel at an elevation of $150 \pm 1$ mm above the bottom of the channel, so as to image a portion

of channel of 40 mm in length from a vertical elevation of 130 mm up to 170 mm. As confirmed by CFD simulations discussed later, at this elevation, the bubbles reached their terminal rise dynamics. Due to the large difference between the densities of water and air (see Table 1), air bubbles in water have very low inertia. Correspondingly, a rise of just a few centimeters is normally sufficient to extinguish the initial transient and reach the terminal dynamics [20,22,23]. Since only one camera was used, the bubbles were characterized using their planar projections, as seen in individual frames. Even though this is normally considered to be sufficiently accurate [22], multiple cameras could clearly provide a more faithful bubble characterization. The error associated with using only one camera in the present case was estimated with CFD simulations, as discussed later.

As illustrated in Figure 2, the image processing methodology adopted here is articulated in four steps:

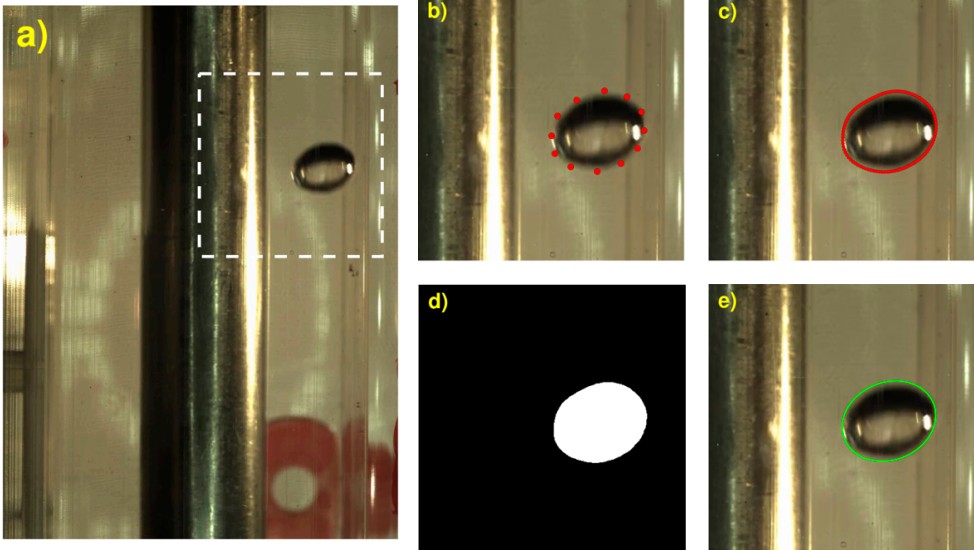

**Figure 2.** Image processing methodology adopted here: (**a**) raw RGB image (the region delimited with the broken line is highlighted in panels b-e), (**b**) raw RGB image detail with discrete points (in red color) manually digitized along the bubble border, (**c**) raw RGB image detail with increased density points (in red color) generated with cubic spline interpolation along the bubble border, (**d**) binary image of the bubble generated from the polygon representing the bubble border, (**e**) raw RGB image detail with equivalent ellipse superimposed (in green color).

1.  Discrete points are manually digitized along the bubble border (Figure 2b);
2.  Cubic spline interpolation is used to increase the density of the points along the bubble border (Figure 2c);
3.  The polygon representing the bubble border is converted to a binary image of the bubble (Figure 2d);
4.  Size and shape of the bubble are computed.

Operatively, the manual location of the points along the bubble border was accomplished with the free Web-based tool WebPlotDigitizer [24], whilst the other image processing steps were carried out with the free software GNU Octave version 4.2.2 [25], relying exclusively on built-in features for the sake of reproducibility (particularly the functions *poly2mask* and *regionprops* of the GNU Octave package 'image'). Manual digitizing was performed on the raw image by visually locating the bubble interface (note the rather sharp intensity variation across the bubble border in Figure 2a) and by manually digitizing discrete points (see Figure 2b) along the interface. The number of manually located points necessary for a successful bubble shape reconstruction was determined empirically via trial and error to be in the range of 10–15 (see Figure 2b), whilst an increase via cubic spline interpolation to 100 points (see Figure 2c) was found sufficient to yield a binary image (see Figure 2d) with a reasonably smooth boundary that properly captured the bubble shape.

Following common practice, the size and shape of the bubbles were characterized using the equivalent diameter and the aspect ratio, respectively. In particular, the instantaneous equivalent diameter $d_{eq}$ of the bubble is the diameter of the circle with the same area as the projection of the bubble:

$$d_{eq}(t) = \sqrt{\frac{4\,A(t)}{\pi}}, \tag{1}$$

where $A$ is the area of the bubble projection and $t$ is the time label of the image, i.e., the time the image was recorded.

Different approaches have been used in the literature to compute the aspect ratio of the bubble projection. The method adopted here is based on the equivalent ellipse (built-in within the function *regionprops* of the GNU Octave package 'image'), which is the ellipse that has the same normalized second central moments as the binary image of the projected bubble. This follows common practice in image analysis, where moments are used to describe the shape of image features by measuring the distribution of pixels with respect to the horizontal and vertical directions [26,27]. A representative example is provided in Figure 2e, where the equivalent ellipse (in green color) is superimposed onto the raw image of the bubble. The instantaneous aspect ratio $E$ of the equivalent ellipse is simply defined as the ratio of the lengths of the minor (*b*) and major (*a*) axes of the ellipse:

$$E(t) = \frac{b(t)}{a(t)}. \tag{2}$$

In addition to the aspect ratio, the inclination of the equivalent ellipse was also computed, which corresponds to the angle between the major axis of the ellipse and the horizontal (the angle is positive if oriented counter-clockwise, negative otherwise). Following common practice, the instantaneous bubble rise velocity $V_{rise}$ was computed as the vertical velocity of the bubble centroid as follows:

$$V_{rise}(t) = \frac{z_c(t + \Delta t) - z_c(t)}{\Delta t}, \tag{3}$$

where $z_c$ is the vertical elevation of the centroid of the bubble and $\Delta t$ is the time elapsed between successive frames (note that, strictly speaking, Equation (3) is exact only in the limit of $\Delta t \to 0^+$; in the present case, however, $\Delta t$ is small enough to make the approximation acceptable).

Measuring errors were on the order of 9–10% for the equivalent diameter, 9–10% for the aspect ratio, and 6–7% for the rise velocity; and were estimated by imaging various calibration standards of known shape and size placed at various positions inside the annular channel.

The relevant dimensionless numbers in single bubble dynamics are the Reynolds number *Re*, the Eötvös number *Eo*, the Weber number *We*, and the Morton number *Mo*; these were computed as follows:

$$Re(t) = \frac{\rho_l\,V_{rise}(t)\,d_{eq}(t)}{\mu_l}, \tag{4}$$

$$Eo(t) = \frac{(\rho_l - \rho_g)\,g\,d_{eq}^2(t)}{\sigma}, \tag{5}$$

$$We(t) = \frac{\rho_l\,V_{rise}^2(t)\,d_{eq}(t)}{\sigma}, \tag{6}$$

$$Mo = \frac{g\,(\rho_l - \rho_g)\,\mu_l^4}{\rho_l^2\,\sigma^3}, \tag{7}$$

where $g$ is the acceleration of gravity. Note that the dimensionless numbers listed above are all instantaneous, except for the Morton number, which was constant and equal to

$2.18 \times 10^{-11}$. Mean values of the equivalent diameter, the aspect ratio, the rise velocity, and the dimensionless numbers listed above were computed by averaging across all instantaneous values of each individual bubble.

The present image processing methodology was developed specifically for the present experimental apparatus. In particular, as shown in Figure 3, the bubbles sometimes slide in front of the rod. When this happens, back lighting is not sufficient to resolve the bubble in its entirety and front lighting is also required, as schematically illustrated in Figure 1a. When front lighting is used, the bubble casts a shadow onto the rod, as can be noticed in Figure 3a–c,e). Clearly, the presence of a moving shadow makes the image background variable. Consequently, this makes the standard image processing approach, which is based on background subtraction, image binarization, and edge detection, unfeasible because subtracting a variable background is problematic. As is evident in Figure 3, the present image processing methodology is functional despite the variable background. Note that painting the rod to mitigate the bubble shadowing was not feasible because of the pool boiling experiments mentioned in Section 2.1, which required a metallic rod with clean surface. Moreover, the present image processing methodology can easily be applied to multiple bubbles which, as explained in Section 2.1, are within the scope of the present experimental setup, whereas resolving multiple bubbles with the standard image processing approach based on background subtraction, image binarization, and edge detection is not straightforward. Within the limits of this study, the present image processing methodology was found to be robust and accurate. In comparison with the standard image processing approach, which is largely computer-based, the present method is clearly more time-consuming. The present image processing methodology is similar to that used for bubble columns by Besagni and Inzoli [28,29], who identified bubbles by manually locating six points along the bubble border, and then calculated an approximating ellipse by directly interpolating through these six manually located points. Though the starting point is clearly the same, the present approach differs from that of Besagni and Inzoli [28,29] in the number of points used to resolve the border of the bubble, and in the way the bubble is reconstructed and its shape characterized.

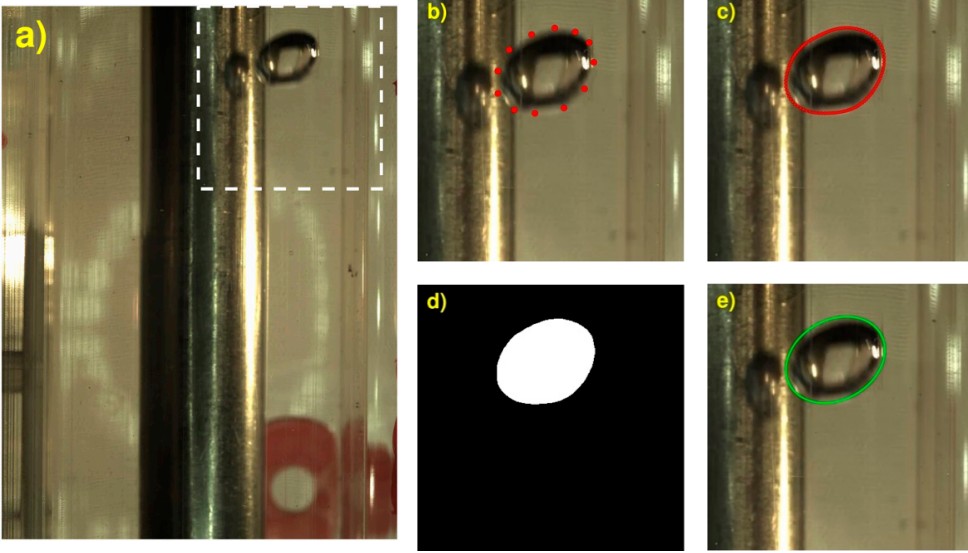

**Figure 3.** Representative example of image processing for a bubble sliding in front of the rod: (**a**) raw RGB image (the region delimited with the broken line is highlighted in panels (**b**–**e**)), (**b**) raw RGB image detail with discrete points (in red color) manually digitized along the bubble border, (**c**) raw RGB image detail with increased density points (in red color) generated with cubic spline interpolation along the bubble border, (**d**) binary image of the bubble generated from the polygon representing the bubble border, (**e**) raw RGB image detail with equivalent ellipse superimposed (in green color).

## 3. Numerical Model

### 3.1. Governing Equations and Discretization Methods

The rise of air bubbles in water was simulated using the open-source package Open-FOAM v.1812 and the Volume of Fluid (VoF) method [30] implemented in the two-phase solver *interIsoFoam* [31]. According to VoF, liquid and gas are treated as a single-mixture fluid and the volume fraction $\alpha$ identifies the fraction of cell volume occupied by a selected primary phase, so that $0 \leq \alpha \leq 1$. *interIsoFoam* solves the unsteady volume fraction, continuity and momentum equations for an isothermal, incompressible two-phase flow and Newtonian fluid in the following form:

$$\frac{\partial \alpha}{\partial t} + \nabla \cdot (\alpha \mathbf{u}) = 0, \tag{8}$$

$$\nabla \cdot \mathbf{u} = 0, \tag{9}$$

$$\frac{\partial (\rho \mathbf{u})}{\partial t} + \nabla \cdot (\rho \mathbf{u} \mathbf{u}) = -\nabla p + \nabla \cdot \left[ \mu \left( \nabla \mathbf{u} + \nabla \mathbf{u}^T \right) \right] + \rho \mathbf{g} + \sigma \kappa \nabla \alpha, \tag{10}$$

where $t$ is time, $\mathbf{u}$ is the fluid velocity vector, $\rho$ and $\mu$ are the mixture fluid density and dynamic viscosity, respectively, $p$ is the pressure, and $\sigma \kappa \nabla \alpha$ introduces the surface tension force estimated via the Continuum Surface Force method [32], with $\sigma$ being the surface tension coefficient and $\kappa$ the local interface curvature, here calculated as $\kappa = \nabla \cdot (\nabla \alpha / |\nabla \alpha|)$.

OpenFOAM discretizes the transport equations above with a finite volume method on a collocated grid arrangement, where all variables are defined at the control volume centres. *interIsoFoam* is a geometric VoF solver which discretizes Equation (8) according to a two-step procedure. First, an interface reconstruction step finds an approximation of the liquid–gas interface within each cell cut by the interface (where $0 < \alpha < 1$), by appropriate isosurface calculations. Then, an interface advection step calculates the volume of fluid crossing each control volume boundary and constituting the convective term of Equation (8), under the assumption that the interface translates steadily across the control volume face during the time interval. Details of the algorithm are provided by Roenby et al. [31]. Unlike OpenFOAM's algebraic VoF solver *interFoam*, *interIsoFoam* guarantees a sharp interface representation, without the need of any artificial interface compression strategy. A first-order, bounded, implicit scheme (*Euler*) is used for the temporal discretization of the flow equations. Second-order schemes are adopted for all spatial derivatives: *Gauss limitedLinearV* and *Gauss vanLeer* for the convective terms in the momentum and volume fraction equations, respectively; *Gauss linear corrected* for all Laplacian schemes and surface normal gradients. OpenFOAM's PIMPLE algorithm (combination of SIMPLE and PISO) is used for pressure–velocity coupling, with 3 PISO correctors (*nCorrectors 3*), no momentum prediction (*momentumPredictor no*), and 2 non-orthogonal correctors (*nNonOrthogonalCorrectors 2*) to account for the utilized non-orthogonal mesh. The residuals thresholds for the iterative solution of the flow equations are set to $10^{-7}$ for the velocity and $10^{-8}$ for both volume fraction and pressure. The time step of the simulation is variable and is calculated based on a maximum allowed Courant number of 0.5.

The present study covers values of the bubble Reynolds number, defined in Equation (4), on the order of $Re \approx 10^3$, and thus turbulence is likely to be present in the wake of the bubbles. RANS and LES turbulence models utilize empirical constants and wall functions calibrated with single-phase flow data, and therefore their applicability to interface-resolved simulations of partially confined bubbles, rising in an otherwise stagnant liquid, is not guaranteed. As such, the numerical results presented in this work were obtained by solving Equations (8)–(10) without any turbulence model, as done in previous studies with similar Reynolds numbers [33–36]. The spatial and temporal resolution of the simulation, chosen upon a grid independence analysis whose results are outlined in Section 3.2 below, set the smallest scales of the vortices that can be fully resolved by the numerical model.

### 3.2. Geometry and Boundary Conditions

The geometry simulated corresponds to the annular channel used for the experiments and was a vertical box of external dimensions $25.3 \times 25.3 \times 350$ mm$^3$, with a coaxial cylindrical rod of diameter of 9.99 mm subtracted from it; a sketch of the entire simulated domain is provided in Figure 4a.

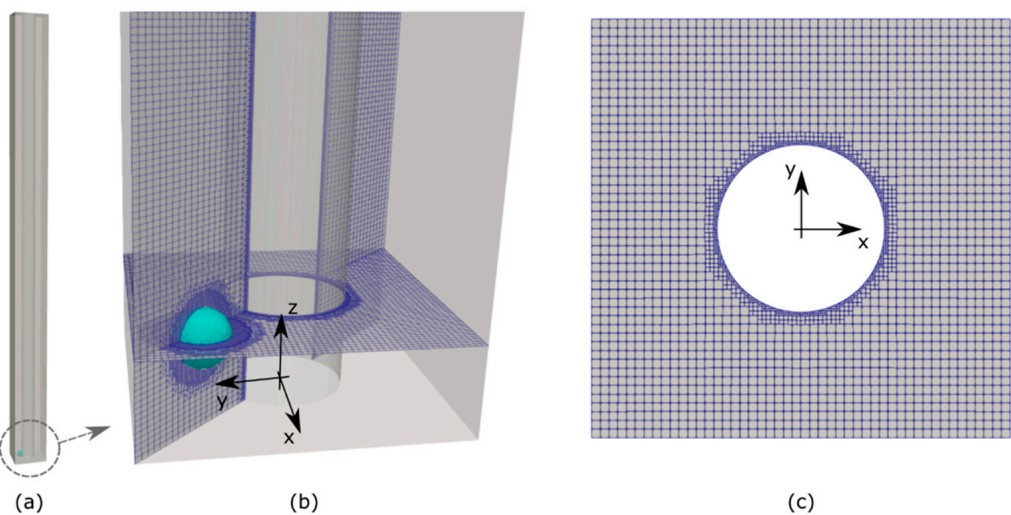

(a)       (b)           (c)

**Figure 4.** Sketch of the computational domain: (**a**) Image of the whole geometry, in slight transparency, with the initial spherical bubble represented in light blue; (**b**) close-up view of bubble and mesh on selected planes at $t = 0$; (**c**) two-dimensional view of the mesh on the channel horizontal cross-section. This setup corresponds to the case with bubble diameter $D_b = 4.5$ mm and mesh with 40 cells per bubble diameter at the liquid–gas interface and 10 in the bulk liquid, which is the mesh used throughout this study.

In order to describe the flow, we adopt a Cartesian reference frame where $z$ denotes the vertical coordinate, with $z = 0$ being the bottom surface of the box, while $x$ and $y$ indicate horizontal axes, with $x = y = 0$ coinciding with the rod axis, and being aligned with the external edges of the box cross-section (see Figure 4b,c). All the domain boundaries except for the top boundary are set as walls, with a no-slip condition for the velocity, a zero-gradient condition for the pressure, and a zero-contact-angle condition for the volume fraction, which models a hydrophilic wall to prevent bubble adhesion. The top wall is set as an outlet boundary, with a constant pressure value and zero-gradient conditions for velocity and volume fraction. The gravitational force is introduced as $\mathbf{g} = -g\mathbf{z}$, with $g = 9.81$ m/s$^2$. The fluids simulated are air and water at 295 K, with properties as shown in Table 1. At time $t = 0$, the computational domain is filled with stagnant water and an initially spherical air bubble is patched nearby the channel bottom, centered at $z = 0.007$ mm and $x = y = 0.0075$ mm, i.e., along the diagonal of the channel cross-section and about halfway between the rod and external box surface; see illustration in Figure 4b.

Adaptive mesh refinement was utilized in order to provide sufficient resolution to the flow, while maintaining a coarse mesh far from the bubble. The background mesh is composed of cubic hexahedra (see Figure 4b,c) with two recursive levels of refinement nearby the rod surface, where the control volumes are clipped to fit the cylindrical surface. The mesh is dynamically refined at the bubble interface during runtime up to the maximum level of refinements selected, and the same criterion is also applied to the initial mesh at time $t = 0$, as can be observed in Figure 4b. The requirements in terms of mesh resolution for the rise of bubbles in stagnant liquid is usually expressed in cells per bubble diameter. Hua et al. [33], Dijkhuizen et al. [34], and Roghair et al. [35] performed direct numerical simulations of air bubbles rising in an infinite pool of stagnant water for Reynolds numbers up to $10^3$ [34,35] and $10^4$ [33], employing 20 cells per bubble diameter. Loisy et al. [37], Balcázar et al. [38], and Esmaeli and Tryggvason [39] studied the rise of bubbles in the

spherical and ellipsoidal regime for Reynolds numbers up to 100 with 25–30 cells per bubble diameter. Cano-Lozano et al. [40], Tripathi et al. [41], and Gumulya et al. [42] analyzed bubble shapes and trajectories for Reynolds numbers up to 100 employing adaptive mesh refinement with resolutions of 128 [40], 82 [41] and up to 220 [42] cells per bubble diameter at the liquid–gas interface. Gumulya et al. [36] simulated the rise of air bubbles in stagnant water for Reynolds numbers up to $10^3$ using adaptive mesh refinement with up to 220 cells/diameter at the bubble interface.

In the present work, numerical simulations are run for a range of bubble diameters $D_b = 3$–6 mm, and Reynolds numbers on the order of $10^3$ are expected. We performed a mesh convergence analysis for a reference $D_b = 4.5$ mm case, employing 5 cells/diameter in the bulk flow and 20 (2 recursive refinements), 40 (3), 80 (4) cells/diameter at the bubble interface, and another arrangement with 10 cells/diameter in the bulk flow and 40 (2 refinements) at the interface. The mesh is updated at the end of every time step. The results are reported in Figure 5. For all the configurations studied, the bubble trajectory (see Figure 5a) is approximately zigzag planar during the first $20D_b$ of the rise, and then develops into a helix of diameter of about 5 mm and pitch 30–35 mm. Throughout the rise, the bubble remains in the quarter of the channel where it was first generated. The velocity of the bubble centroid, calculated as $V_c = |d\mathbf{x_c}|/dt$ with $\mathbf{x_c} = (x_c, y_c, z_c)$ being the centroid position, is reported in Figure 5b. In all cases, the bubble speed oscillates between $V_c = 0.17$ m/s and 0.25 m/s, with local minima occurring when the bubble approaches the walls. The time-average speed, calculated between $z_c = 0.2$–0.3 m, is in the range $V_c = 0.216$–0.22 m/s for all the meshes tested, indicating less than 2% differences. The vertical component of the rise velocity, estimated as $V_{rise} = dz_c/dt$, shows time averages in the range $V_{rise} = 0.199$–0.205 m/s for all the meshes tested, which compare well with the experimental value. As such, all the meshes tested exhibited similar results and the configuration with 10–40 cells per bubble diameter in the bulk bubble was adopted for the simulations presented in this work. This corresponded to a total of about 5 million mesh cells for the reference case with $D_b = 4.5$ mm.

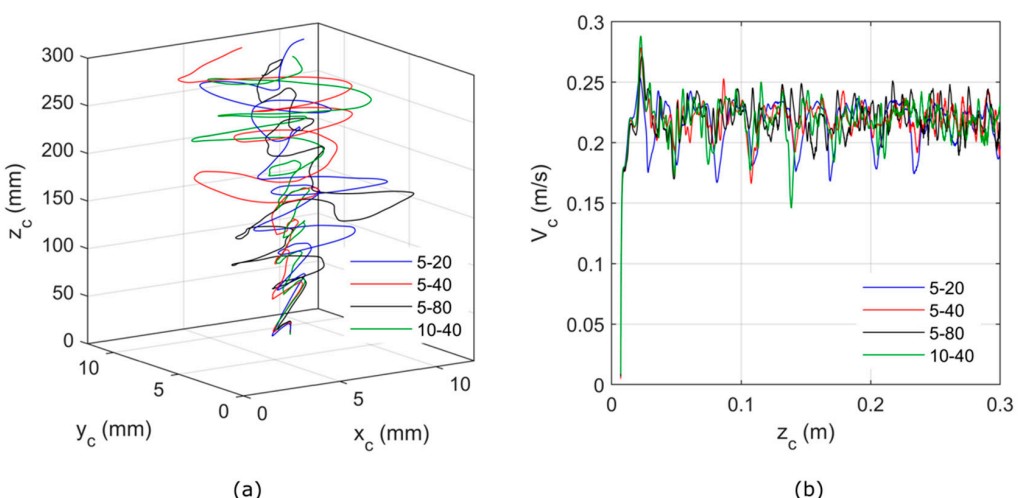

(a)　　　　　　　　　　　　　　　　　　　　　　　(b)

**Figure 5.** Results of grid independence analysis; $D_b = 4.5$ mm. (**a**) Position and (**b**) speed of the bubble centroid during the rise. The legend in (**a,b**) indicates the number of cells per bubble diameter in the bulk and at the bubble interface. The average speed of the bubble, calculated between $z_c = 0.2$–0.3 m, is: 5–20) 0.216 m/s, 5–40) 0.218 m/s, 5–80) 0.22 m/s, 10–40) 0.217 m/s.

A time step dependency analysis was carried out by testing a Courant number of 0.1: no appreciable differences were observed from the 0.5 Courant number case. Note that for a Courant number of 0.5, the simulation time step oscillated around $\Delta t = 3.5 \times 10^{-5}$ s (and about $\Delta t = 0.7 \times 10^{-5}$ s for a Courant of 0.1), which is well within the capillary time

step limit, $\Delta t = \left[ (\rho_l + \rho_g) h^3 / (4\pi\sigma) \right]^{1/2} \cong 4 \times 10^{-5}$ s, with $h = D_b/40$ being the mesh size across the liquid–gas interface.

Further preliminary tests (not documented here) were conducted by starting the simulation with $g = 0$, and increasing $g$ gradually as the time elapsed (at a rate of 1 m/s$^2$ every 0.1 s) until its actual value was achieved, in order to emulate a gradual initial rise of the bubble with a smaller initial deformation. However, once the nominal value of $g$ was restored, the bubble shape and speed showed no differences with the dynamics obtained by setting $g = 9.81$ m/s$^2$ from $t = 0$. This showed that the impulsive bubble start adopted in the simulations did not affect the subsequent bubble rise dynamics. If this was the case, then the actual growth and detachment of the bubble would need to be simulated, adding considerable complexity to the numerical model.

### 3.3. Postprocessing of Numerical Data

From the numerical simulations, bubble interface data are extracted during runtime with regular temporal frequency, typically every 0.005 s, ensuring about 300 frames for each run. Figure 6a shows a snapshot of the bubble during the rise for the $D_b = 4.5$ mm case, run with the coarsest (5–20) mesh. The points representing the liquid–gas interface are identified as the $\alpha = 0.5$ iso-surface. This list of points is read in Matlab (version R2018b) and the built-in function *alphaShape* is utilized to create a bounding volume enveloping these points, enabling the calculation of geometrical quantities such as bubble surface area and volume. Further topological queries such as the extraction of centers, nodes and normal vectors of the boundary facets can be addressed with the function *triangulation*, as illustrated in Figure 6b. In order to characterize the geometry of the bubble, this is modelled using two-dimensional ellipses and three-dimensional ellipsoids. From the list of interface points, the bubble projections on the $xz$- and $yz$-planes are first obtained. Then, for each projection the boundary nodes are extracted and utilized to fit a two-dimensional ellipse [43] as shown in Figure 6c,d. From the fit, the ellipse axes lengths and aspect ratio are calculated. A different function is used to fit a three-dimensional ellipsoid [44] to the interface points and calculate axes lengths and aspect ratios. This procedure is repeated for all the saved bubble interface data, to finely resolve the bubble dynamics over time.

### 3.4. Validation of the Numerical Model

The numerical model was first validated by comparison with the experimental data of Bhaga and Weber [45] for air bubbles rising in stagnant, unconfined, aqueous sugar solutions. Eight different sets of conditions were selected from Bhaga and Weber [45] (see Figure 7) covering values of the Reynolds number from close to unity to the largest value achieved in their work, $Re = 259$. The simulations were run in a three-dimensional domain of size $12D_b \times 12D_b \times 32D_b$, with slip boundary conditions applied to all boundaries. Adaptive mesh refinement was employed, with 10 cells/diameter in the bulk flow. For the cases illustrated in Figure 7a–c,e 40 (2 recursive refinements) cells/diameter were employed at the bubble interface, whereas, for the cases illustrated in Figure 7d,f–h, 80 (3 recursive refinements) cells/diameter were used at the bubble interface to better capture the thin bubble skirt. The Courant number of the simulations was limited to 0.025. The bubble was initialized at the domain center, near the bottom wall. The terminal bubble shapes for simulations and experiments are illustrated in Figure 7. The bubble rose following a vertical rectilinear path at all conditions. Six of these cases (Figure 7a–d,f,g) are characterized by very similar values of the Eötvös number, $Eo = 114$–$116$, and Morton numbers ranging from $Mo = 848$ to $8 \times 10^{-4}$, while three of these (Figure 7e,g,h) have similar Morton numbers, $Mo \approx 8 \times 10^{-4}$, and an Eötvös number ranging from $Eo = 32$ to $237$. As the Morton number is decreased or the Eötvös number is increased, the bubble rises faster and shape transitions from a dimpled ellipsoidal-cap to a flattened spherical cap, eventually exhibiting an open wake and an asymmetric shape as the Reynolds number approaches 150. There is qualitatively a good agreement between the numerical and experimental bubble shapes reported in Figure 7.

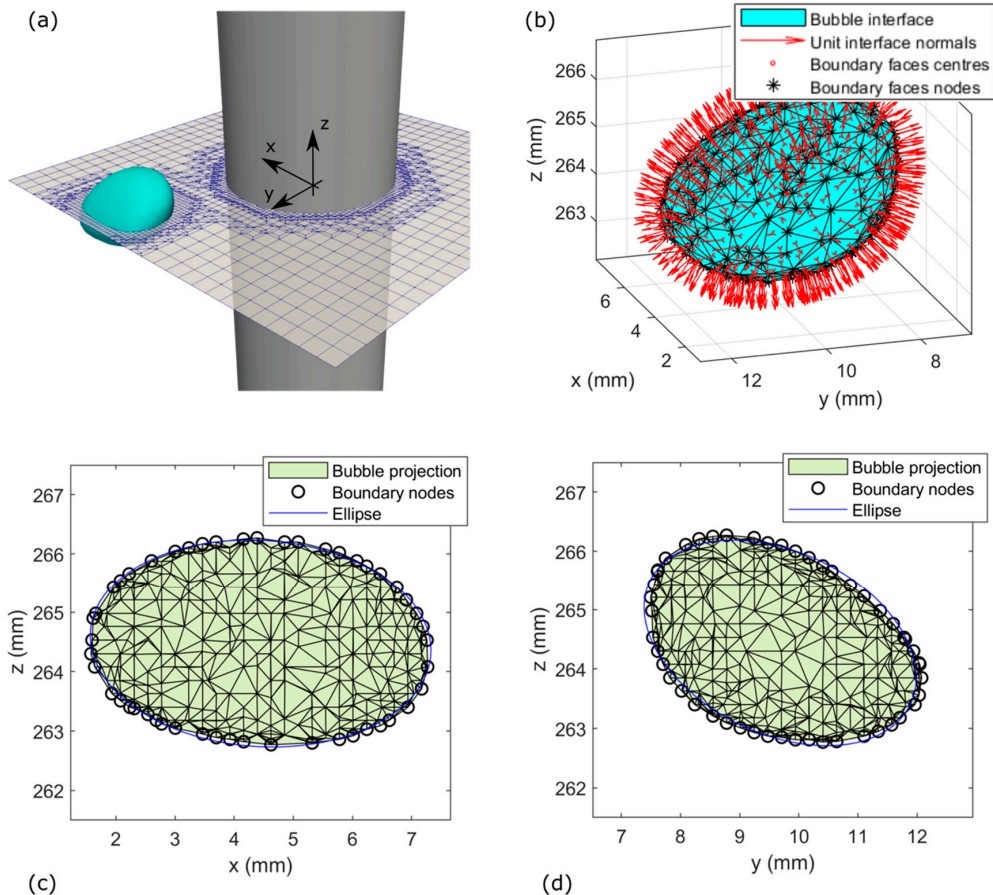

**Figure 6.** Postprocessing of numerical simulation data; $D_b = 4.5$ mm. (**a**) Snapshot of the bubble surface (in blue) identified as $\alpha = 0.5$ iso-surface, with the central rod (dark gray) and the computational mesh over a selected horizontal plane (in slight transparency); (**b**) Reconstruction of the bubble surface in Matlab; (**c**,**d**) Projected bubble on a (**c**) $xz$-plane and (**d**) $yz$-plane, with identified boundary nodes of the projection and two-dimensional fitted ellipse. For representation purposes, the results shown in this figure were obtained with a coarser mesh, with 20 and 5 cells per bubble diameter at the interface and bulk liquid, respectively.

A quantitative comparison of terminal bubble speed and aspect ratio is offered in Figure 7i–l. The bubble speed predicted by the simulation for $Re \leq 100$ is always within 5% of the experimental data, whereas there is a systematic tendency to underestimate the experimental data for larger Reynolds numbers, although the maximum deviation remains below 10%. The same tendency of numerical simulations to underestimate the bubble velocities measured by Bhaga and Weber at larger Reynolds numbers was previously reported by the numerical studies of Hua and Lou [46] and Gumulya et al. [42], who used different simulations techniques and observed up to 10% deviations from the experimental data. The aspect ratio of the bubbles depicted in Figure 7l was calculated in the numerical simulations by taking the ratio between the height and width of the projection of the bubble profile on the $xz$ and $yz$ planes, thus disregarding the indentation at the bubble bottom. As the Reynolds number increases, the bubble flattens and the aspect ratio decreases from $E \approx 0.7$ when $Re \approx 2$ to $E \approx 0.2$ when $Re > 100$. The agreement between numerical and experimental data is excellent, with deviations below 5%, which is less than the uncertainty of the experimentally measured data for ellipsoidal bubbles indicated by Bhaga and Weber [45].

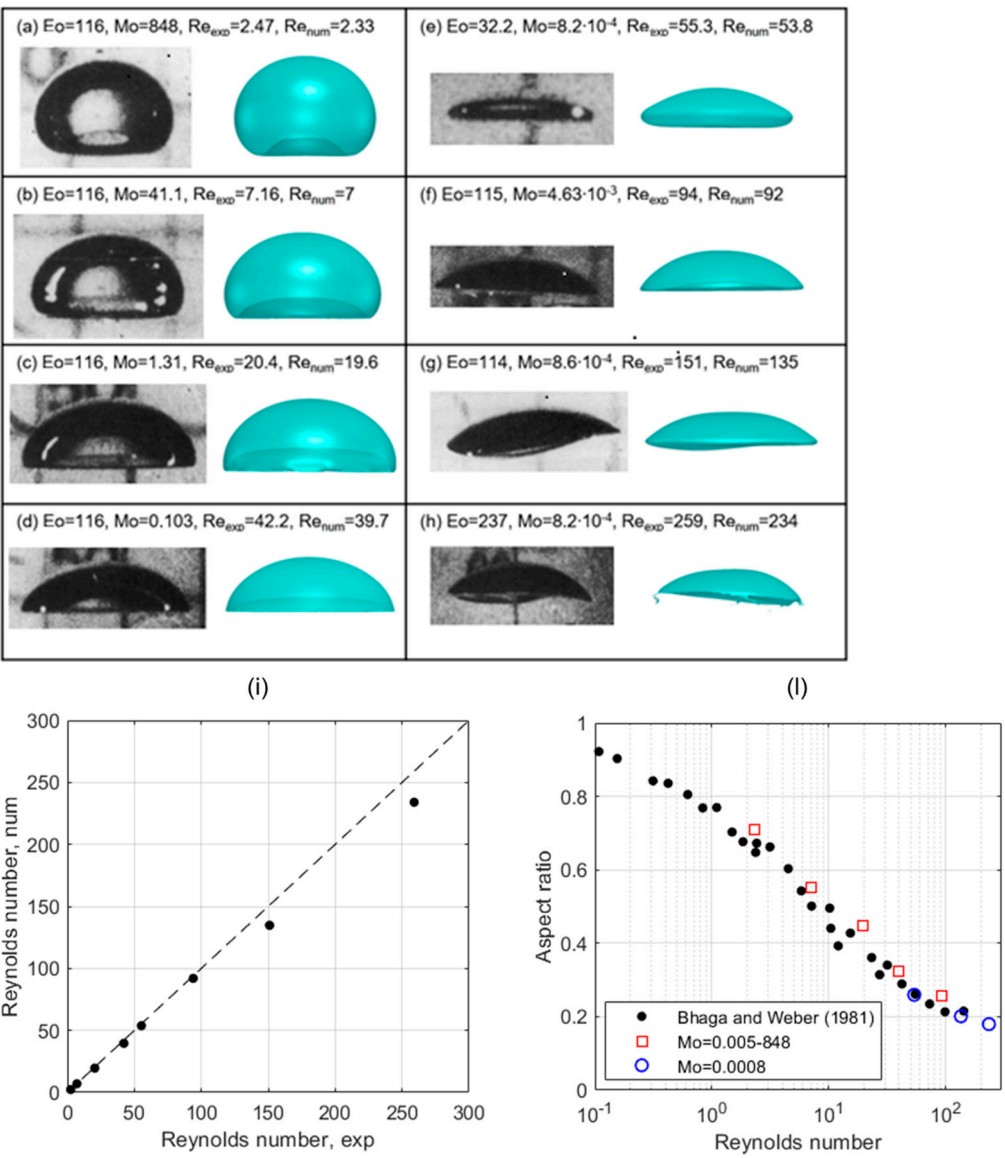

**Figure 7.** Validation of the numerical model versus the experimental data of Bhaga and Weber [45]. (**a**–**h**) Experimental and numerical terminal bubble shapes at different conditions, sorted by increasing values of the Reynolds number; (**i**) Numerical versus experimental bubble Reynolds number and (**l**) aspect ratio, with the red squares in (**l**) identifying simulation results for cases (**a**–**d**,**f**), and blue circles identifying simulation results for cases (**e**,**g**,**h**).

The second validation test that was performed consists of the simulation of an air bubble rising in an unconfined pool of stagnant water, for a range of bubble diameters $D_b = 3$–5.2 mm, which covers the range analyzed in confined conditions in the Results section. The experimental data for unconfined air–water systems of Tomiyama et al. [17] and Veldhuis [47] are utilized as benchmark for the numerical simulations. Tomiyama et al. [17] studied trajectories, shapes and velocities of air bubbles rising in stagnant water for a range of bubble diameters $D_b = 0.5$–5.5 mm, and reported very different bubble dynamics, depending on the initial shape deformation of the bubble at the instant of release from the injection nozzle. Bubbles with small initial shape deformation rose with a zigzag motion, lower speed and a larger aspect ratio; as opposed to bubbles released with a larger initial shape deformation, which rose with a helical motion, larger speed and lower aspect ratio. Importantly, bubbles released with larger initial deformation exhibited a significant scattering on their terminal speed and shape, as shown in Figure 8. Veldhuis [47] studied the behavior of air bubbles rising in water at similar conditions to Tomiyama et al. [17],

with bubbles released from a capillary tube. They observed different bubble trajectories in the range $D_b$ = 3–5.2 mm, from zigzag to spiral and, eventually, chaotic as the bubble size increased. Their data for bubble rise velocities and aspect ratio sit at the boundary of the dataset from Tomiyama et al. [17], see Figure 8, with faster and more flattened bubbles. This can be ascribed to the large initial shape deformation, which might be induced by the injection capillary. Figure 8 also includes the rise velocity prediction extracted from an empirical correlation proposed by Park et al. [48], which applies to air bubbles rising in water in the range $D_b$ = 0.1–20 mm; these predictions fall in between Tomiyama et al. [17] and Veldhuis [47] measurements. Numerical simulations were run in a box of size $28D_b \times 28D_b \times 150D_b$. The fluid properties of air and water at 293 K were considered. A spherical bubble was initialized at the domain center, near the bottom wall. The domain was meshed using an adaptive mesh with 40 cells per bubble diameter at the air–water interface and 10 cells/diameter far from it. Meshes with higher refinement at the interface and/or in the bulk flow did not yield appreciable differences in the results. The Courant number of the simulations was set to 0.1. In the range of bubble diameters simulated, the bubbles rose with paths in between zigzag and helical. The terminal bubble rise speed and aspect ratio were calculated as time averages of the instantaneous values extracted for $z_c > 30D_b$, when the rolling mean of the bubble speed became constant. The results from the simulations are presented in Figure 8. Bubble rise speed and aspect ratio sit within the area of the graphs occupied by the dataset of Tomiyama et al. [17] for helically rising bubbles. Although mild, the numerical data confirm the experimentally observed trends that both bubble rise speed and aspect ratio increase as the bubbles size is reduced and the effect of surface tension forces becomes more significant.

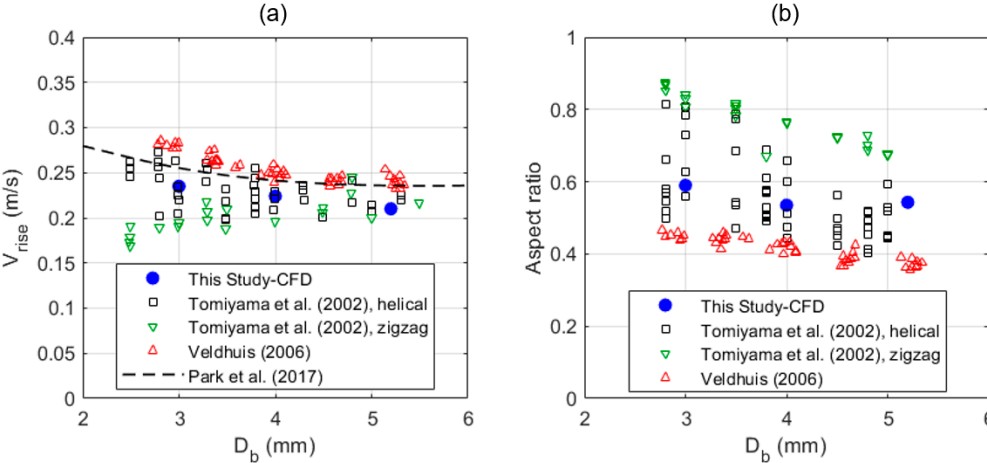

**Figure 8.** Validation of the numerical model versus the experimental data of Tomiyama et al. [17] and Veldhuis [47]: (**a**) bubble rise velocity and (**b**) aspect ratio versus bubble diameter. The black dashed line in (**a**) indicates the bubble rise speed predicted using Park et al. [48] correlation.

In summary, validation simulations of air bubbles rising in a liquid pool were conducted, covering a range of Reynolds numbers $Re \approx$ 2–1100. Bubble rise velocities and aspect ratios agreed well with the selected experimental benchmark data, thus confirming the validity of the numerical model utilized in this work.

## 4. Results and Discussion

During the experiments we recorded and analyzed 30 individual bubbles, all generated under seemingly identical operating conditions, corresponding to a total of 597 individual frames, which should be sufficient to provide a reliable bubble shape distribution (approximately 300 or more frames are needed to provide reliable bubble size/shape distributions, as discussed by Besagni and Inzoli [28]). To help interpret the measurements and generalize the experimental observations to bubble diameters in the range of 3–6 mm, we numerically simulated the rise of six individual bubbles with diameters of 3 mm, 3.5 mm, 4 mm, 4.5 mm,

5.2 mm, and 6 mm. The mean values of the main bubbles' parameters from the present experiments (averaged over all of the 30 bubbles recorded) and numerical simulations are summarized in Table 2.

**Table 2.** Mean values of the main bubbles' parameters.

| | EXP | CFD | | | | | |
|---|---|---|---|---|---|---|---|
| $d_{eq}$ (mm) * | 4.3 | 3.0 | 3.5 | 4.0 | 4.5 | 5.2 | 6.0 |
| $V_{rise}$ (m/s) | 0.18 | 0.216 | 0.214 | 0.210 | 0.200 | 0.199 | 0.190 |
| $E$ | 0.81 | 0.679 | 0.626 | 0.602 | 0.605 | 0.608 | 0.634 |
| $Eo$ | 2.49 | 1.22 | 1.66 | 2.17 | 2.75 | 3.67 | 4.88 |
| $We$ | 1.93 | 1.94 | 2.21 | 2.45 | 2.50 | 2.85 | 3.00 |
| $Re$ | 805 | 678 | 782 | 880 | 944 | 1083 | 1194 |

(*) In the experiments $d_{eq}$ was deduced from the area of the bubble planar projection, whilst in the numerical simulations it was deduced from the actual bubble volume.

As explained later in Section 4.1, the bubble mean equivalent diameter deduced from the two-dimensional bubble projection in the experiments is 5–6% smaller than the actual mean equivalent diameter deduced from the bubble volume in the numerical simulations. The mean bubble diameter of 4.3 mm measured in the experiments, therefore, corresponds to an actual bubble diameter of about 4.5 mm, so that the experimental figures summarized in Table 2 can be compared directly with the CFD simulations for the bubble of 4.5 mm. When comparing the present measurements with the numerical simulations, it can be noted that the simulated bubble rises about 10% faster than the measured bubble, and its aspect ratio is about 30% lower. Though not significant, the difference is larger than the present measuring errors, particularly for the aspect ratio (measuring errors are 6–7% for the rise velocity and 9–10% for the aspect ratio, as noted previously). The discrepancy between the present experiments and numerical simulations can be traced back to two main reasons: (1) turbulence in the wake of the bubbles that is not captured in the numerical simulations and (2) a possible contamination in the test water, as explained below.

At the Reynolds numbers covered in the present simulations (678–1194), turbulence is likely to be present in the wake of the bubbles. We performed preliminary tests by using a k-ω SST turbulence model [49]; however, the bubble exhibited a perfectly planar zig-zag trajectory throughout its rise, with an even larger speed. Therefore, the results presented in this work were obtained without any turbulence model, which means that turbulent eddies with spatial and temporal scales smaller than the mesh and time step size of the simulation were not captured. Normally, in external flows, the emergence of turbulence is accompanied by an increase in the drag in comparison with the laminar flow. Therefore, as a consequence of not accounting for turbulence, the numerical model can be expected to somewhat underpredict the drag, and thus overpredict the rise velocity. A higher rise velocity would increase the pressure force acting on the bubble which, in turn, would flatten, thereby yielding a lower aspect ratio.

Aside from turbulence, water contamination can be a cofactor responsible for the mismatch between the present experiments and numerical simulations. As is well known, fully eliminating contamination and surfactants from water is particularly difficult, if at all possible [3]. Therefore, despite the fact that we employed nominally micro-filtered and high-purity water, a minor contamination in the test water cannot be excluded. Clearly, the water in the numerical simulations is perfectly pure. Since bubbles rising through clean liquids rise faster and are more deformed than the corresponding bubbles rising in contaminated systems [1], a minor contamination in the test water could explain the lower rise velocity and the less pronounced bubble deformation observed in the present experiments, in comparison with the numerical simulations.

Despite the limitations in the present experiments and numerical simulations, the present results are nonetheless informative for the single bubble dynamics in the confined annular channel considered here.

### 4.1. Preliminary CFD Results

Preliminary CFD results that support the design of the present test system and of the associated measuring procedure are presented below.

As explained previously, in the experiments the bubbles were generated by blowing air through a capillary pipe at a frequency of one bubble generated every 2.2 s (bubbling frequency of 0.45 Hz). The CFD results provided in Figure 9 confirm that the delay of 2.2 s is sufficient for the liquid to recover between successive bubble passages. In particular, Figure 9a depicts a control volume of coordinates $x > 0$, $y > 0$, $0.0115 \text{ m} < z < 0.016 \text{ m}$, corresponding to a 4.5 mm long slice of a quarter of the annular channel located immediately above the initial position of the bubble, whereas the time variation of the specific kinetic energy of the water contained in this control volume is presented in Figure 9b,c. It is evident that, as soon as the bubble enters the control volume, the kinetic energy of the water increases as a consequence of the agitation induced by the rising bubble. Once the bubble leaves the control volume, the kinetic energy begins to decrease, indicating that viscous dissipation is gradually damping the water motion. By the time the bubble reaches the top of the channel at time 1.5 s, the kinetic energy of the water inside the control volume decreased by about four orders of magnitude, and, correspondingly, the mean velocity (Figure 9d,e) decreased by about two orders of magnitude, as compared to the peak values reached when the bubble rose through the control volume. Therefore, by the time the bubble reaches the top of the channel, the motion of the water in the control volume is practically extinguished. This confirms that the air flow rate set-point used in the experiments ensures continuous bubbling at a steady rate with a delay between successive bubbles sufficient to allow liquid recovery, and thus treat each bubble as a single bubble rising through a practically stagnant liquid.

Predicted bubble rise velocities are presented versus vertical elevation in Figure 10a. As can be noted, the initial acceleration transient is rather fast and ends after a rise of a few centimeters. This confirms that, by the time they reach the vertical elevation where the camera is positioned, the bubbles have reached their terminal dynamics. This latter state is clearly not a steady-state because the predicted rise velocity is fluctuating, as can be noted in Figure 10a and further discussed in the following Section 4.2.

As explained previously, only one camera was used in the experiments and the bubbles were characterized using their planar projections as seen in individual frames. The error associated with using only one camera in the present case is on the order of a few percent. This is illustrated in the CFD results presented in Figure 10b which provides, for all simulated bubbles, the ratio of the mean equivalent diameter deduced from the two-dimensional bubble projection (the diameter of the circle with the same area as the projection of the bubble) to the three-dimensional mean equivalent diameter (the diameter of the sphere with the same volume as the bubble). As noted previously, in the present case the variation of the air density due to the hydrostatic pressure variation along the channel is negligible. Consequently, the volume of the bubbles and the three-dimensional mean equivalent diameter are constants. As can be noted in Figure 10b, the mean equivalent diameter deduced from the two-dimensional bubble projection is 5–6% smaller than the three-dimensional mean equivalent diameter, an error that is rather small and within the present experimental resolution (as previously mentioned, the measuring error for the equivalent diameter is on the order of 9–10%).

### 4.2. Instantaneous Bubble Dynamics

The experimental instantaneous dynamic of one representative bubble is documented in Figure 11, where the sequence of the bubble projection perimeters extracted from consecutive images is provided together with the corresponding instantaneous variations of the bubble equivalent diameter, rise velocity, aspect ratio and inclination of the equivalent ellipse, as well as Reynolds, Eötvös and Weber number values. The bubble rise trajectory indicates a zigzag motion of small amplitude, and the instantaneous variations of the bubble parameters clearly indicate that the bubble rise dynamic is wobbling, as all bubble

parameters show quite pronounced variations. This is also confirmed by the CFD results documented in Figure 10a, which clearly show a non-steady-state terminal dynamic.

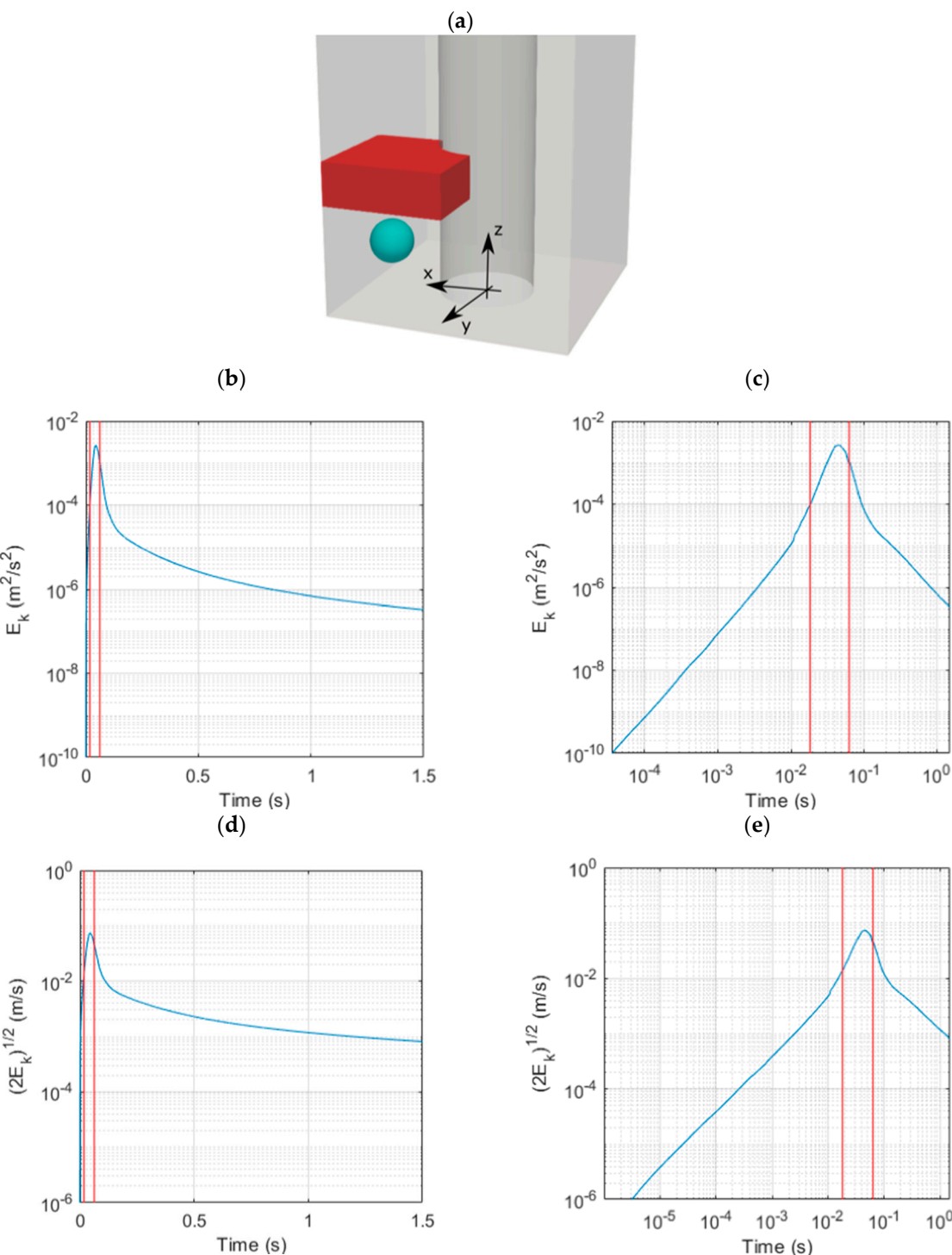

**Figure 9.** (a) View of the control volume (in red) used in the calculation of the kinetic energy of the water during bubble rise ($D_b = 4.5$ mm), with the blue sphere depicting the bubble at $t = 0$; time variation of the kinetic energy of the water inside the control volume in linear (**b**) and logarithmic (**c**) scale; time variation of the mean velocity (computed as the square root of the kinetic energy) of the water inside the control volume in linear (**d**) and logarithmic (**e**) scale. The red vertical lines mark the times the bubble enters (leftmost red line) and leaves (rightmost red line) the control volume.

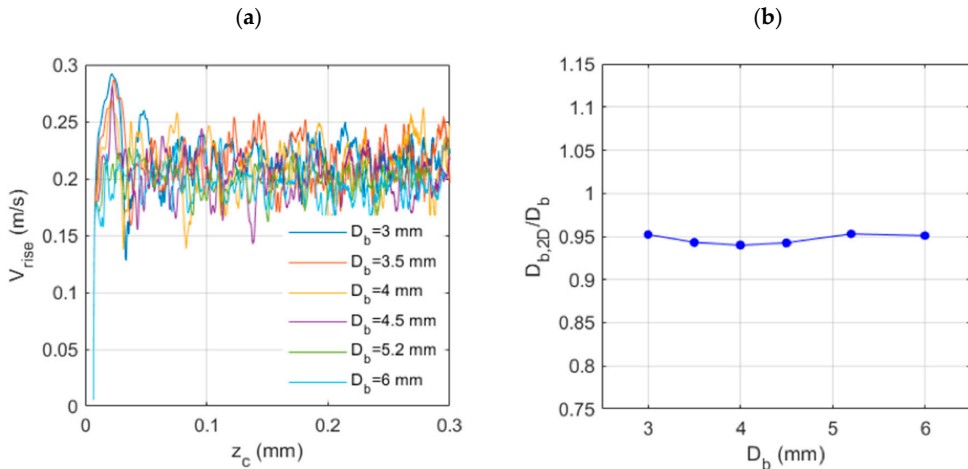

**Figure 10.** Predicted bubble rise velocities plotted versus vertical elevation (**a**), and ratio of the mean equivalent diameter deduced from the two-dimensional bubble projection to the three-dimensional mean equivalent diameter deduced from the bubble volume (**b**).

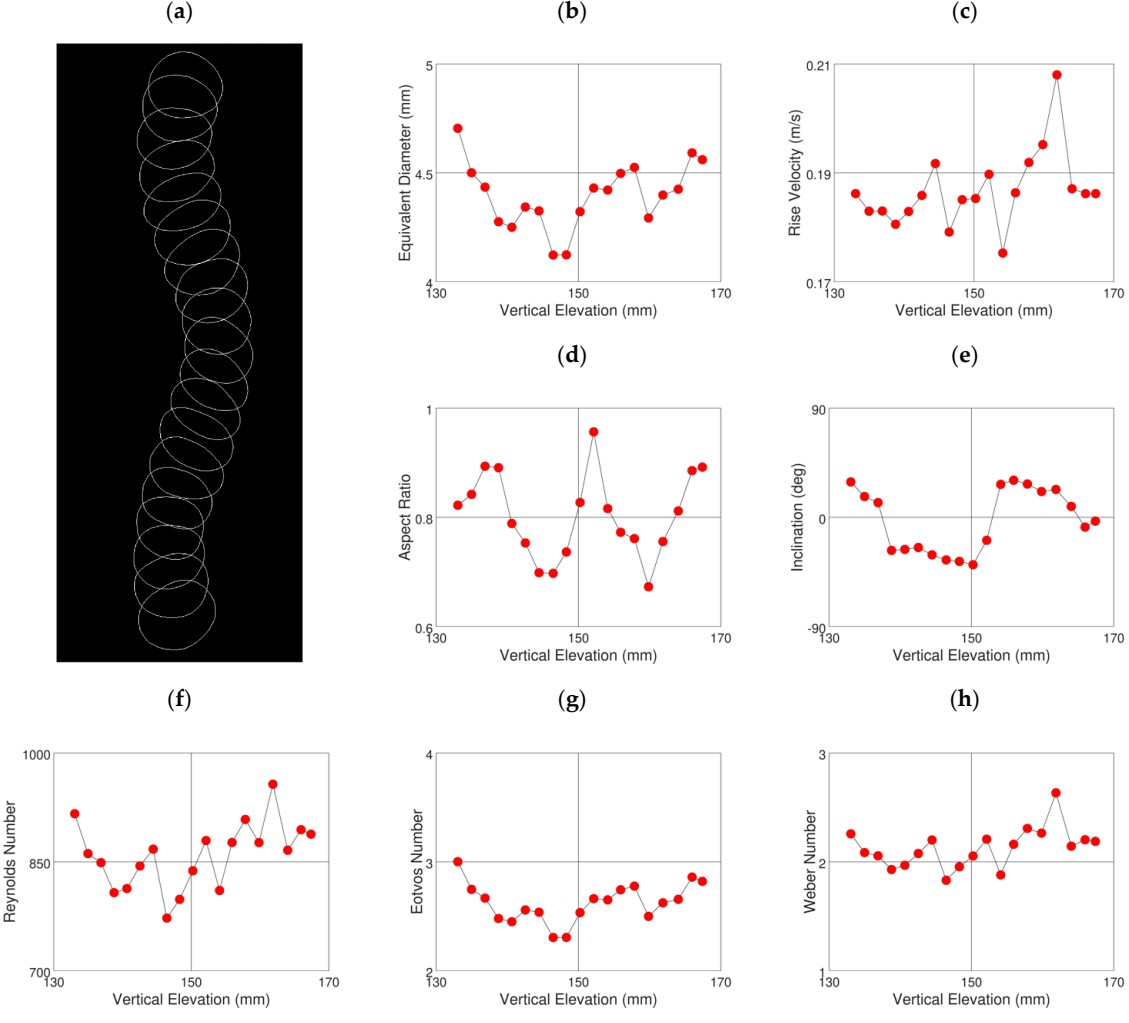

**Figure 11.** Instantaneous dynamic of one representative measured bubble: (**a**) sequence of the bubble projection perimeters extracted from consecutive images (the time elapsed between successive frames is 10.36 ms) and corresponding instantaneous variations of the bubble equivalent diameter (**b**), rise velocity (**c**), aspect ratio (**d**) and inclination (**e**) of the equivalent ellipse, and Reynolds (**f**), Eötvös (**g**) and Weber (**h**) number values.

The experimental instantaneous dynamic of all the measured bubbles is documented in Figure 12, where the rise trajectories of the centroid of the bubble projections extracted from consecutive images are provided together with the histograms of the corresponding instantaneous variations of the bubble equivalent diameter, rise velocity, aspect ratio and inclination of the equivalent ellipse, and Reynolds, Eötvös and Weber number values. Even though the instantaneous dynamic of each single bubble is wobbling (see Figure 11), the various trajectories appear reasonably similar, suggesting a fairly repeatable bubble rise dynamic. In turn, this suggests that the initial deformation of the bubbles did not vary appreciably during the experiments.

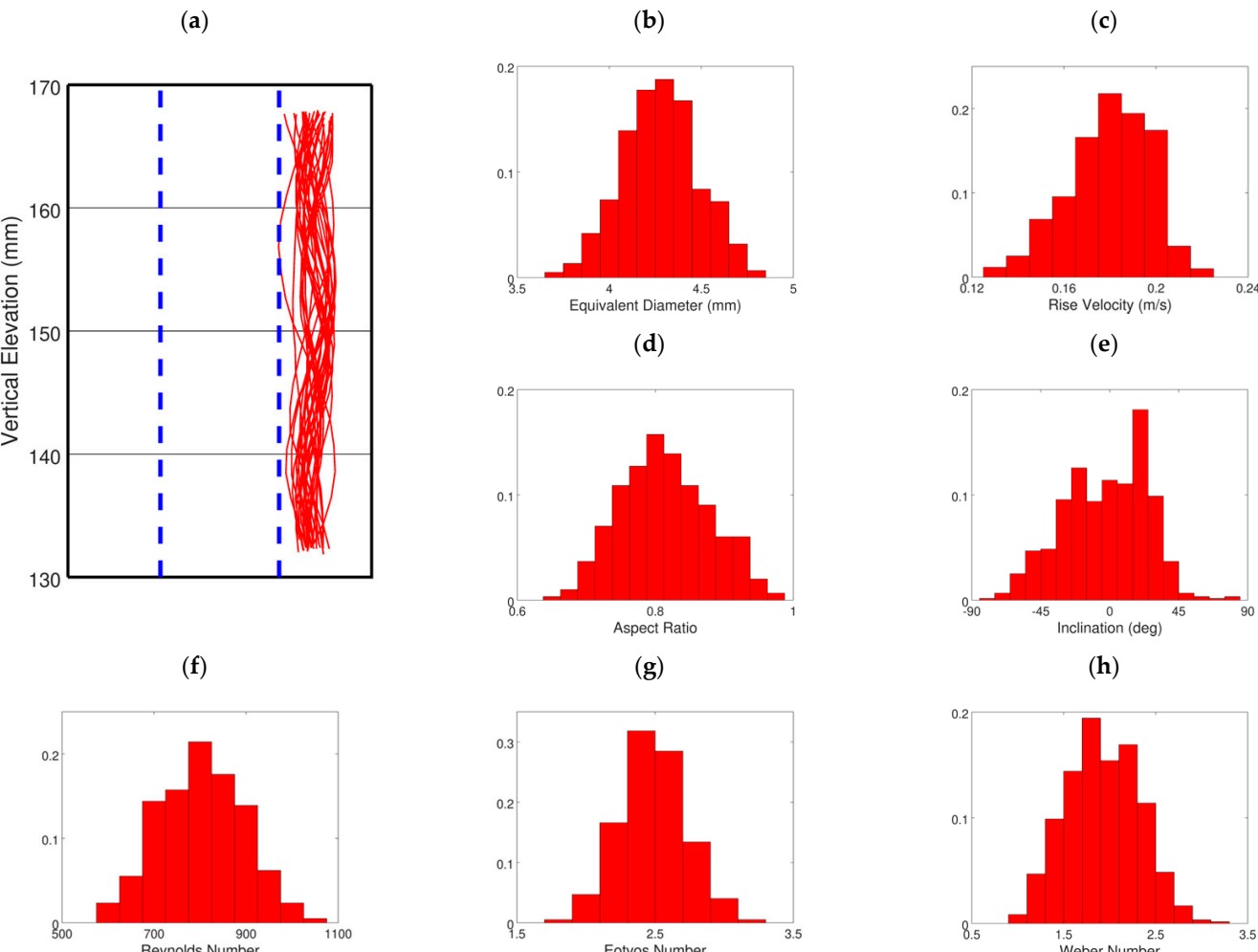

**Figure 12.** Instantaneous dynamic of all measured bubbles: (**a**) rise trajectories of the centroid of the bubble projections extracted from consecutive images (the vertical dashed lines in blue color denote the position of the rod), and histograms of the corresponding instantaneous variations of the bubble equivalent diameter (**b**), rise velocity (**c**), aspect ratio (**d**) and inclination (**e**) of the equivalent ellipse, and Reynolds (**f**), Eötvös (**g**) and Weber (**h**) number values.

Air bubbles of a few mm in size rising in stagnant unconfined water at ambient conditions have an ellipsoidal shape and exhibit a wobbling rise dynamic [1,50]. The present results show a wobbling rise dynamic, and therefore indicate that the confinement of the annular channel does not change the qualitative character of the bubble dynamic.

### 4.3. Bubbles Mean Shape

The mean shape results of the bubbles are presented in Figure 13, where the mean aspect ratio from the present measurements and CFD simulations is presented as a function of the mean Eötvös number (panel a), mean Weber number (panel b), mean Tadaki

number *Ta* (panel c), and mean Reynolds number (panel d). The Tadaki number is defined as follows:

$$Ta = Re\,Mo^{0.23}, \tag{11}$$

where *Re* and *Mo* are the mean Reynolds and Morton numbers, respectively. The non-circular confined channel data provided by Tomiyama et al. [15] are also included in Figure 13, whereas the non-circular confined channel data provided by Venkateswararao et al. [14] are not included since the authors measured the terminal rise velocity but did not measure the shape of the bubbles.

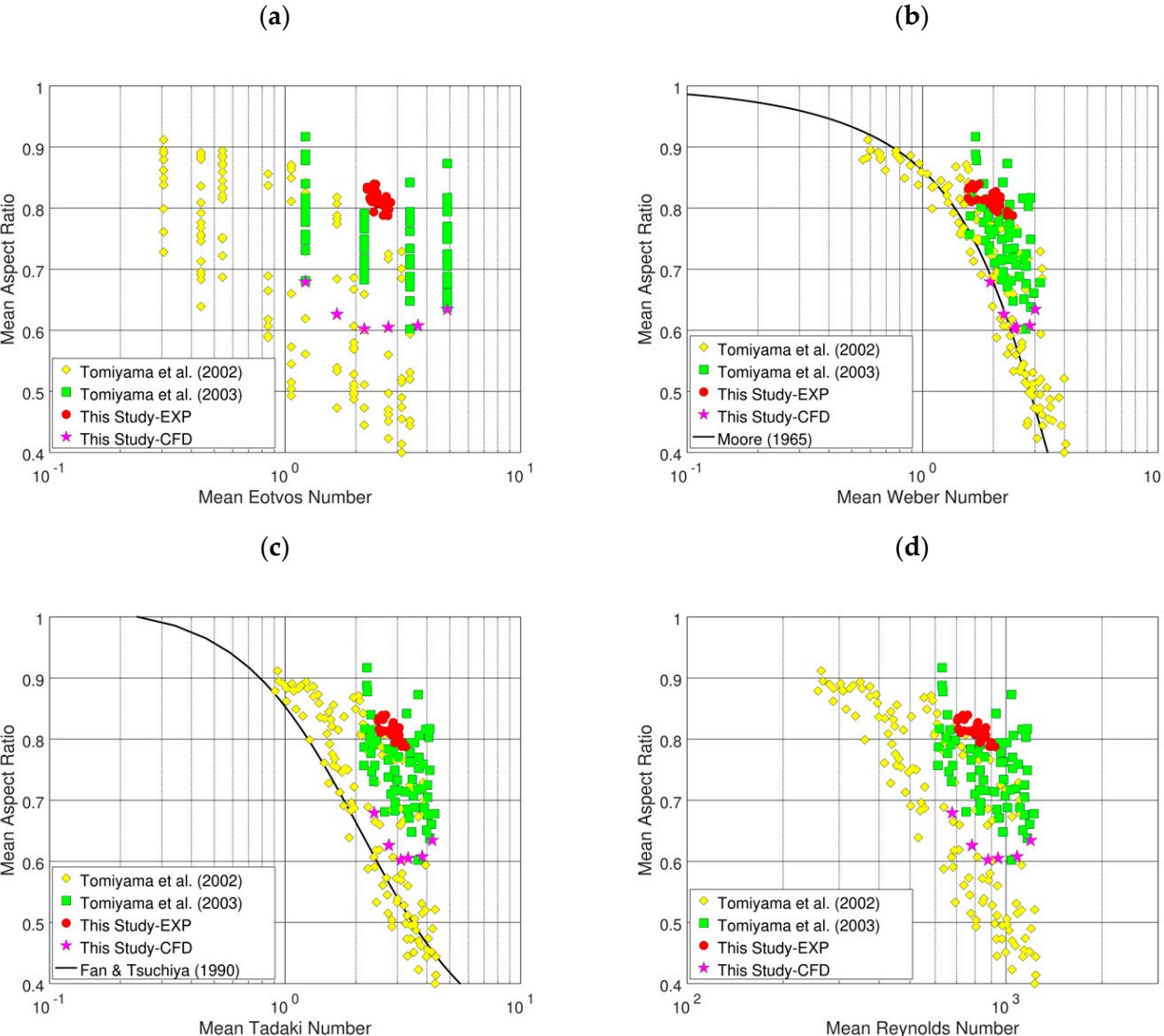

**Figure 13.** Mean aspect ratio: measured and computed values versus Eötvös number (**a**), Weber number (**b**), Tadaki number (**c**), and Reynolds number (**d**) (Tomiyama et al. (2002) [17]: unconfined rise; Tomiyama et al. (2003) [15] and this study: confined rise).

To better put the non-circular confined channel results into perspective, the unconfined rise data by Tomiyama et al. [17] are also included in Figure 13: these refer to air bubbles at ambient conditions rising through unconfined water, and are therefore informative of the shapes of air bubbles rising through water when confinement effects are not present. Note that all data included in Figure 13 refer to air bubbles rising through clean water at ambient conditions, so that the main difference between individual data subsets is the presence/absence of confinement effects. Moreover, being only a function of the thermo-

physical properties of the fluids (see Equation (7)), the Morton number has the same constant value for all data in Figure 13, so that the Tadaki number in the present case is essentially the Reynolds number multiplied by a constant. This is why the present aspect ratio data in Figure 13 show similar clustering when plotted versus the Tadaki number (panel c) and the Reynolds number (panel d).

The aspect ratio prediction methods proposed by Moore [51], Equations (12) and (13), and by Fan and Tsuchiya [52] (as quoted in [20]), for bubbles rising through unconfined clean liquids are also included for comparison in Figure 13 (panes b and c, respectively):

$$E = \left(1 + 0.1406\, We - 0.0089\, We^2 + 0.0287\, We^3\right)^{-1}, \tag{12}$$

$$E = \{0.77 + 0.24\tanh[1.9(0.4 - log_{10}Ta)]\}^2 \quad 0.3 < Ta < 20. \tag{13}$$

In its original formulation, the prediction method proposed by Moore [51] is an implicit relation for the aspect ratio as a function of Weber number. The relation in Equation (12) used here is a convenient explicit approximation developed by Loth [3] for moderate deformations ($E \gtrsim 0.5$). Finally, the present data are compared, in the parity plot in Figure 14, with the predictions of the aspect ratio prediction method proposed by Loth [3]:

$$E = 1 - (1 - E_{min})\, tanh(c_E We) \quad 0.2 < Re < 5000, \tag{14}$$

$$E_{min} = 0.25 + 0.55\, exp(-0.09\, Re) \tag{15}$$

$$c_E = 0.165 + 0.55\, exp(-0.3\, Re). \tag{16}$$

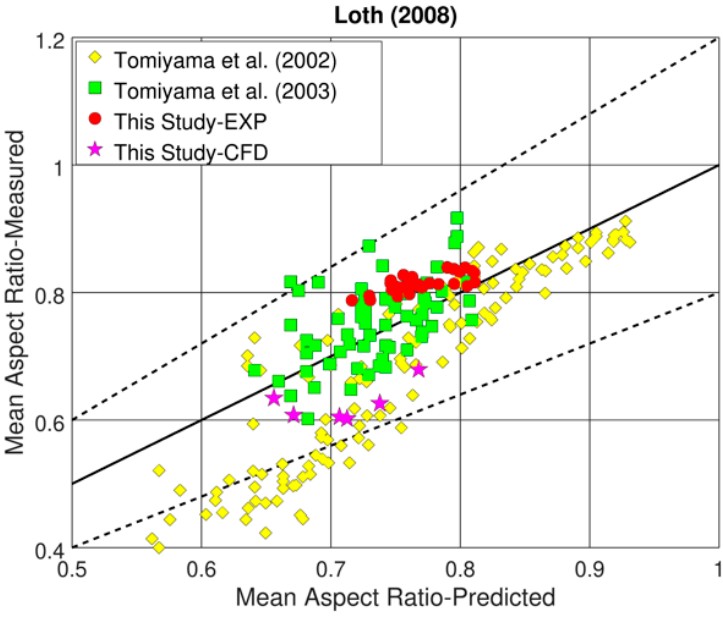

**Figure 14.** Mean aspect ratio: present data vs. predictions of the Loth [3] method (the dashed lines are ±20% bounds).

As can be noted in Figure 13, the Weber number appears to be the dimensionless group that is most effective at clustering the present data. The Tadaki number is also effective, though not as effective as the Weber number, whereas the Eötvös number does not appear to be effective at clustering the present data. Similar findings concerning the clustering effectiveness of the Weber, Tadaki and Eötvös numbers were reported by Liu et al. [20] for air bubbles rising through unconfined water and glycerol aqueous solution. The clustering effectiveness of the Weber number suggests that the shape of the present bubbles is largely controlled by the competition between the fluid-dynamic force acting on the bubble surface, which causes deformation, and the surface tension force, which resists deformation. Viscous

dissipation in the liquid also plays a role, as evident from the relatively successful clustering effectiveness of the Tadaki and Reynolds numbers. The usefulness of the Weber number for predicting the mean aspect ratio is also indirectly confirmed in Figure 14, where it can be noted that the Loth [3] prediction method fits the present data fairly well. Even though this prediction method formally depends on both the Weber number and the Reynolds number, the effect of the latter is restricted to low and moderate values of the Reynolds number: indicatively below about 100. Above this limit, which happens to be the case for the present data (as can be noted in Figure 13d), the expressions in Equations (15) and (16) reached their asymptotic limiting values and, correspondingly, the Loth [3] prediction method essentially depends only on the Weber number. As can be noted in Figure 14, this prediction method seems more effective at predicting the confined rise data, as opposed to the unconfined data, part of which are overpredicted. The available data are however too restricted in scope to draw any definite conclusions.

By comparing the aspect ratios in Figure 13 for the data generated with non-circular confined channels (present experiment and Tomiyama et al. [15]) to the unconfined rise data (Tomiyama et al. [17]) and prediction methods it seems clear that, for a given Weber, Tadaki or Reynolds number value, the aspect ratio of the bubbles rising though a confined channel is higher than the aspect ratio of the corresponding bubbles rising through unconfined water. This indicates that the effect of the confinement is to reduce the deformation of the bubbles and, correspondingly, increase the mean aspect ratio.

As can be noted in Table 2, all dimensionless numbers for the simulated bubbles scale in direct proportion to the size of the bubble. It follows that the CFD data points in Figure 13 are oriented left-to-right in proportion to the size of the bubble: the leftmost point refers to the smallest bubble (3 mm), and the rightmost point refers to the largest bubble (6 mm). The present CFD results in Figure 13b,c show a change in trend with increasing Weber or Tadaki number. For small bubble size (3–4 mm), the aspect ratio decreases with the increasing Weber or Tadaki number. This agrees with the trend for an unconfined rise: in fact, the prediction methods in Equations (12) and (13) indicate that, when the bubbles are rising through unconfined liquids, the aspect ratio decreases as the Weber or Tadaki number increase. On the other hand, for larger bubbles (4.5 mm and above) the aspect ratio levels off and then increases with the increasing Weber or Tadaki number, which is the opposite of what happens during unconfined rise. This suggests that, in the present non-circular annular channel, the effect of the confinement on the shape of the bubble scales with the bubble size, which is the same qualitative trend observed with bubbles rising through confined circular tubes (Clift et al. [1], Krishna et al. [6]).

### 4.4. Bubble Mean Rise Velocity

The mean bubble rise velocities from the present measurements and CFD simulations are presented as a function of the mean bubble diameter in Figure 15a, together with the data for non-circular confined channels provided by Venkateswararao et al. [14] and Tomiyama et al. [15].

To better put the non-circular confined channel results into perspective, the unconfined rise data of Tomiyama at al. [17] are also included in Figure 15a, together with the prediction method developed by Mendelson [53] for bubbles rising through unconfined liquids:

$$V_{rise} = \sqrt{\frac{2\,\sigma}{\rho_l\,d_{eq}} + 0.5\,g\,d_{eq}};\tag{17}$$

The data are also presented in Figure 15b as the mean drag coefficient versus mean Reynolds number. Following common practice, the mean drag coefficient $C_D$ and mean rise velocity were linked through a steady-state force balance between drag and buoyancy:

$$C_D\,\frac{1}{2}\rho_l V_{rise}^2\,\frac{\pi}{4}d_{eq}^2 = (\rho_l - \rho_g)\,g\,\frac{\pi}{6}d_{eq}^3;\tag{18}$$

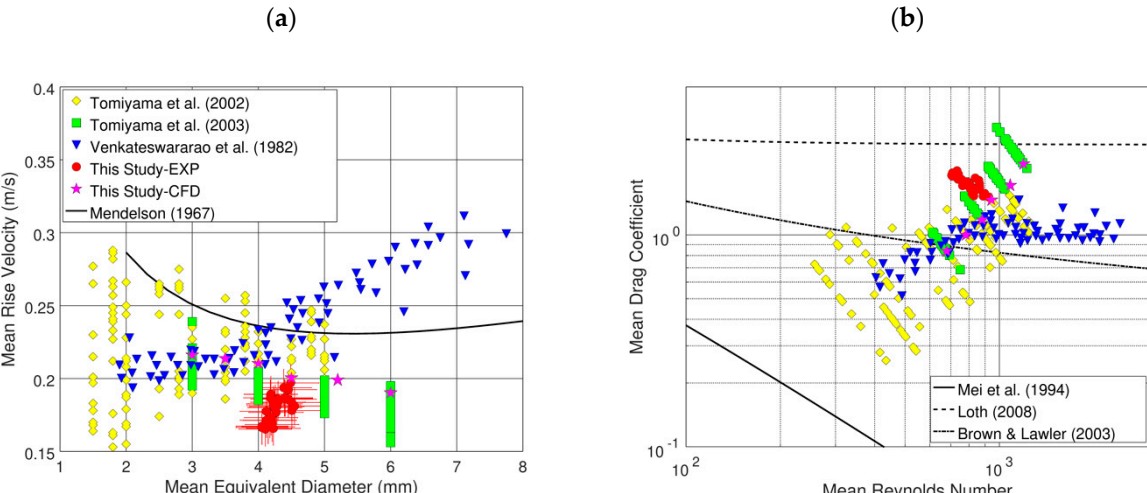

**Figure 15.** Mean rise velocity versus bubble mean diameter (**a**) and mean drag coefficient versus mean Reynolds number (**b**) (Tomiyama et al. (2002) [17]: unconfined rise; Tomiyama et al. (2003) [15], Venkateswararao et al. (1982) [14], and this study: confined rise).

From Equation (18), the mean drag coefficient can be computed from the mean rise velocity, and vice versa:

$$C_D = \frac{4\,(\rho_l - \rho_g)\,g\,d_{eq}}{3\,\rho_l V_{rise}^2}, \quad V_{rise} = \sqrt{\frac{4\,(\rho_l - \rho_g)\,g\,d_{eq}}{3\,\rho_l\,C_D}}. \tag{19}$$

The prediction method proposed by Mei et al. [54] for spherical bubbles rising through clean unconfined liquids, Equation (20), and the prediction method proposed by Loth [3] for spherical-cap bubbles rising through unconfined liquids, Equation (21), are also included in Figure 15b; these provide a lower bound and an upper bound, respectively, for bubbles rising through unconfined liquids:

$$C_D = \frac{16}{Re}\left\{1 + \left[\frac{8}{Re} + \frac{1}{2}\left(1 + \frac{3.315}{\sqrt{Re}}\right)\right]^{-1}\right\}, \tag{20}$$

$$C_D = \frac{8}{3} + \frac{16}{Re}; \tag{21}$$

For further reference, the drag prediction method proposed by Brown and Lawler [55] for solid spheres is also included in Figure 15b:

$$C_D = \frac{24}{Re}\left(1 + 0.150\,Re^{0.681}\right) + \frac{0.407}{1 + \frac{8.710}{Re}}; \tag{22}$$

Note that all data included in Figure 15 refer to air bubbles rising through clean water at ambient conditions, so that the main difference between individual data subsets is the presence/absence of confinement effects.

The present measurements and CFD simulations and the data for non-circular confined channels provided by Venkateswararao et al. [14] and Tomiyama et al. [15] are compared, in Figure 16, with the modified Mendelson prediction method, Equation (23), which is widely used for bubbles rising through confined circular tubes:

$$V_{rise} = \sqrt{\frac{2\,\sigma}{\rho_l\,d_{eq}} + 0.5\,g\,d_{eq}}\left[1 - \left(\frac{d_{eq}}{D}\right)^2\right]^{3/2}, \tag{23}$$

where $D$ is the diameter of the confining circular tube. On the right-hand side of Equation (23), the square root term corresponds to the Mendelson [53] prediction method in Equation (17) for bubbles rising through unconfined liquids, whilst the multiplying term is a correction proposed by Clift et al. [1] that accounts for the confinement. In Figure 16, the velocity from measurements or CFD simulations is divided by the unconfined rise velocity predicted from Equation (17), and then the velocity ratio is plotted as a function of the diameter ratio $d_{eq}/D$. For the present measurements and CFD simulations the hydraulic diameter was used in place of the tube diameter $D$, whilst for the data of Venkateswararao et al. [14] and of Tomiyama et al. [15] the inner subchannel hydraulic diameter was used (as carried out by Tomiyama et al. [15]). The solid line in Figure 16 corresponds to the above confinement correction by Clift et al. [1]. The data by Krishna et al. [6], which refer to air bubbles at ambient conditions rising through confined water in circular tubes, are also included in Figure 16 for comparison with the non-circular channel data.

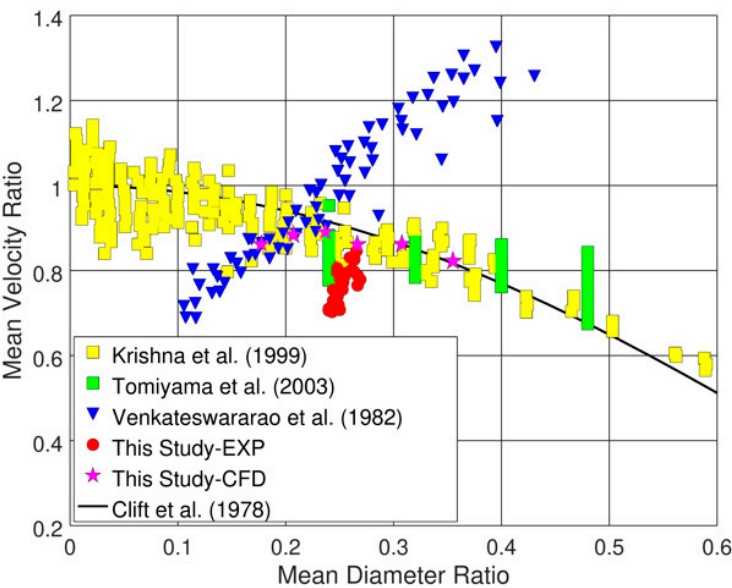

**Figure 16.** Velocity ratio versus diameter ratio.

Finally, the present measurements and CFD simulations, together with the data of Venkateswararao et al. (1982) [14] and Tomiyama et al. (2003) [15], are compared in the parity plots in Figure 17, with the prediction methods proposed by Tomiyama et al. [56], Equation (24); Tomiyama et al. [17], Equation (25); and Rodrigue [57], Equations (26)–(27):

$$C_D = max\left\{min\left[\frac{16}{Re}\left(1 + 0.15\,Re^{0.687}\right), \frac{48}{Re}\right], \frac{8}{3}\frac{Eo}{Eo+4}\right\}, \tag{24}$$

$$V_{rise} = \frac{asin\sqrt{1-E^2} - E\sqrt{1-E^2}}{1-E^2}\sqrt{\frac{8\,\sigma}{\rho_l\,d_{eq}}E^{4/3} + \frac{(\rho_l - \rho_g)\,g\,d_{eq}}{2\,\rho_l}\frac{E^{2/3}}{1-E^2}}, \tag{25}$$

$$V_{rise} = \frac{F}{12}\left(\frac{\rho_l^2\,d_{eq}^2}{\mu_l\,\sigma}\right)^{-1/3}\frac{\left(1 + 1.31\times10^{-5}\,Mo^{11/20}\,F^{73/33}\right)^{21/176}}{\left(1 + 0.020\,F^{10/11}\right)^{10/11}}, \tag{26}$$

$$F = g\left(\frac{\rho_l^5\,d_{eq}^8}{\mu_l^4\,\sigma}\right)^{1/3}. \tag{27}$$

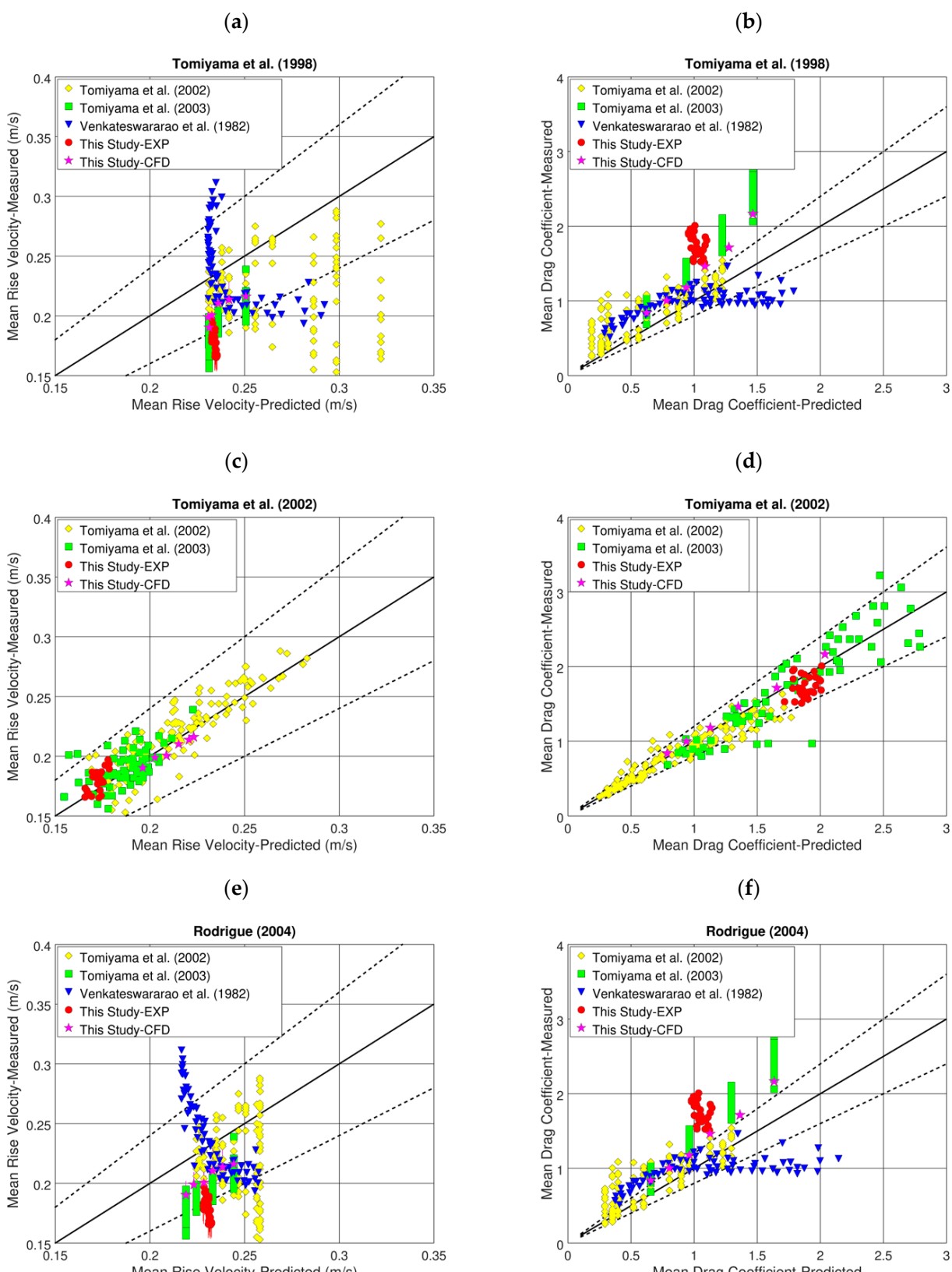

**Figure 17.** Mean rise velocity and corresponding drag coefficient: present data vs. predictions of the Tomiyama et al. [56] method (**a**,**b**), of the Tomiyama et al. [17] method (**c**,**d**), and of the Rodrigue [57] method (**e**,**f**); the dashed lines are ± 20% bounds. (Tomiyama et al. (2002) [17]: unconfined rise; Tomiyama et al. (2003) [15], Venkateswararao et al. (1982) [14] and this study: confined rise).

The rise velocity corresponding to the drag coefficient from Equation (24) and the drag coefficients corresponding to the rise velocities from Equations (25) and (26) were computed according to Equation (19). The data provided by Venkateswararao et al. [14] are not included in Figure 17c,d because the authors did not measure the shape of the bubbles (the bubble aspect ratio is needed as an input to Equation (25)).

As can be noted in Figure 15a, for small bubble size (below about 3–3.5 mm) the rise velocity data generated with non-circular confined channels (present CFD simulations, Venkateswararao et al. [14], and Tomiyama et al. [15]) agree with each other and do not seem to depart significantly from the unconfined rise data (Tomiyama et al. [17]) included for comparison, albeit these latter data are more widely scattered: possibly the consequence of a more pronounced variation of the bubbles' initial deformation during the tests. For larger bubble sizes (above about 3.5 mm), the present experiments and CFD simulations, as well as the data of Tomiyama et al. [15], are in fair agreement with each other and indicate that the rise velocity gradually decreases as the bubble size increases, with values of the rise velocity that appear consistently lower than the unconfined rise data and prediction method. On the other hand, the data of Venkateswararao et al. [14] indicate an opposite trend: the rise velocity gradually increases as the bubble size increases, with values of the rise velocity that appear consistently higher than the unconfined rise data and prediction method.

For the present CFD simulations and for the data of Tomiyama et al. [15], therefore, the effect of the confinement is to reduce the deformation of the bubbles, which increases the mean aspect ratio and reduces the mean rise velocity, which is the same qualitative trend observed with bubbles rising through confined circular tubes [1,6]. On the other hand, according to the data of Venkateswararao et al. [14], the effect of the confinement is an increase in the mean rise velocity. Unfortunately, the authors did not measure the shape of the bubbles, and thus it is not possible to provide a complete assessment of their data. However, it is clear that the cross-section of the test section used by the authors was considerably larger than those used here and used by Tomiyama et al. [15], and this could be the reason for the different trend observed. In turn, this would indicate that, with non-circular channels of a complex shape, the size of the channel cross-section may affect the dynamics of the bubbles, and results generated with comparatively small channels may not extrapolate to channel sizes of industrial relevance.

As noted by Loth [3], Equations (20) and (21) provide a lower bound and an upper bound to the drag of bubbles rising through unconfined liquids, respectively. In other words, drag data fall in the area comprised between the two curves. As can be noted in Figure 15b, this is also the case for most of the available confined rise data. Only a few data points by Tomiyama et al. [15] peak above Equation (21), indicating that the reduction in rise velocity due to confinement can yield a substantial increase in the associated drag coefficient.

As can be noted in Figure 16, the agreement between the present data and CFD simulations, and the data of Tomiyama et al. [15] and Krishna et al. [6] with the modified Mendelson prediction method, is fair but the scatter is substantial. This indicates that the confinement effect may also depend on other factors beyond the diameter ratio that essentially capture the geometry of the system, possibly the Reynolds number, as is the case for the solid spheres falling through confined liquids [55]. However, the available data are too restricted in scope to draw any definite conclusions. It is evident in Figure 16 that the trend of the measurements by Venkateswararao et al. [14] is remarkably different from that of the rest of the data. Nonetheless, it is not possible to rule these data out as outliers: note that the present CFD simulation results for the small bubble size (3–3.5 mm) agree with these data. Moreover, as already noted the cross-section of the test section used by Venkateswararao et al. [14] is considerably larger than those used by the others, and this could explain the different trend observed. It may also be the case that the subchannel hydraulic diameter, which is presently used as a representative length scale, does not fully capture the effect of the channel cross-sectional shape on the dynamics of the bubbles. What

seems clear is that the available data are too restricted in scope to duly assess confinement effects with channels of a complex shape, and more investigations are clearly required.

From the parity plots in Figure 17, it can be noted that the prediction method of Tomiyama et al. [17] is quite successful at fitting both the unconfined and the confined rise data, whereas the other prediction methods considered here are ineffective. As can be noted from inspecting Equations (24)–(27), the prediction method of Tomiyama et al. [17] is the only one that explicitly incorporates the shape of the bubbles via the aspect ratio. This confirms that the size, shape, and the rise velocity of ellipsoidal bubbles are closely linked together, and should all be incorporated into prediction methods.

## 5. Concluding Remarks

Using experiments and numerical simulations, we studied the shape and the rise velocity of single air bubbles, measuring 3–6 mm in diameter, rising through water and confined inside a non-circular annular channel. Our main findings are summarized below:

- The confinement of the present annular channel did not affect the qualitative behavior of the bubbles, which exhibited a wobbling rise dynamic similar to that observed with bubbles rising through unconfined liquids. The effect of the confinement was evident on the shape and rise velocity; the bubbles were less deformed and rose more slowly in comparison with bubbles rising through unconfined liquids;
- The present results are in fair agreement with previous observations by Tomiyama et al. [15] on the shape and rise velocity of air bubbles rising though water, confined in a small cross-section subchannel, and with available data on confined rise through circular tubes. However, the observations by Venkateswararao et al. [14] on air bubbles rising through a larger cross-section tubular test section show an increase in rise velocity as consequence of the confinement, which is a trend that was not observed in other studies on confined rise through smaller channels. This indicates that confinement effects with non-circular channels of complex shape could be more complicated than those observed with circular tubes, and both the size of the channel cross-section and its shape may affect the dynamics of the bubbles; therefore, results generated with comparatively small channels may not extrapolate to channel sizes of industrial relevance. The available data are, however, too restricted in scope to draw any definite conclusions, and more investigations are clearly needed;
- The present data and numerical simulations, as well as the other data collected from the literature and used here, indicate that the size, shape, and rise velocity of ellipsoidal bubbles are closely linked together, and this should be considered when designing prediction methods;
- The image processing methodology developed and used here, based on manually digitizing points along the bubble border, is robust and effective at dealing with the variable image background caused by the bubble shadow. This technique can be extended to multiple bubbles of interest in bubble columns;
- The synergetic use of experiments and numerical simulations proved to be an effective approach for the study of single bubble rise in confined geometries. In particular, we found the numerical simulations instrumental in providing a better insight into the measurements, and effective for generalizing the experimental observations, thereby compensating for the limitations of the test setup. Still, we found the numerical simulations somewhat hampered by the current limitations in RANS turbulence models for bubbly flows, and further validation work in this respect would be particularly beneficial.

**Author Contributions:** Conceptualization, A.C. and M.M.; experiments, A.C.; simulations, M.M.; writing—original draft preparation, A.C. and M.M.; writing—review and editing, A.C. and M.M. All authors have read and agreed to the published version of the manuscript.

**Funding:** This research received no external funding.

**Institutional Review Board Statement:** Not applicable.

**Informed Consent Statement:** Not applicable.

**Data Availability Statement:** All data discussed are directly presented in the paper and are therefore directly accessible.

**Acknowledgments:** M. Magnini acknowledges the use of Athena at HPC Midlands+, which was funded by the EPSRC grant EP/P020232/1, as part of the HPC Midlands+ consortium. M. Magnini acknowledges the support from EPSRC, through the BONSAI (EP/T033398/1) grant.

**Conflicts of Interest:** The authors declare no conflict of interest.

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
