# Peer review of "Shapes and Rise Velocities of Single Bubbles in a Confined Annular Channel: Experiments and Numerical Simulations"

_fluids, doi:10.3390/fluids6120437_

Round 1
Reviewer 1 Report
This manuscript studied the bubble rising velocity and aspect ratio of single bubbles in a confined channel. Experimental setups, procedures, and data postprocessing methods are well demonstrated although the experiment was limited. Numerical simulations ranging from different conditions are performed and detailed comparisons/validations with literature are presented. Although experiment results are not well fit with simulations due to experimental errors claimed by the authors, this study revealed some facts/trends worth publication. I have a few comments for the authors as below:
- On line 104, the author mentioned current experiment is a 'compromise' of requirements. However, I didn't see Point 2 and Point 3 are mentioned elsewhere in the manuscript. The author should mention ongoing/future experiments (if any) in Point 2 and Point 3, or else an experimental design that is more tuned towards Point 1, like using a transparent rod should be adopted.
- Figure 1 is difficult to understand. The fillings of inclined lines for water in Figure1a should be removed. There should be some labeling on each part, like a rod, acrylic wall, and water level (horizontal cyan lines in Figure1c).
- The author used manual marks for locating bubbles. The marks are generated at pixel level? It's better to provide how this was performed, for example, by looking at the intensity difference in a group or by looking at the gradient map and finding the maximum gradient pixel location. It's mentioned Besagni and Inzoli used the same method, do they have the same marking criteria excluding the number of mark points?
- Line 400, should be ∆t = 3.5 x 10 -5. A lot of similar errors at other places.
- Line 410-414, what's the point of mentioning this test? It looks like not relevant to the topic of the manuscript.
- Figure 7, the Reynolds number used in Figure7d, 7e, 7h for experiments could not be found in Figure 7l. Why Reynolds numbers are so different in g and h, is it possible to tune the CFD result to make them closer?
Reviewer 2 Report
In this paper, the authors developed both experimental and numerical methods to quantitatively study the shapes and rise velocities of single air bubbles rising through stagnant water confined inside an annular channel. The structure and logic of this article is clear and sound. The image processing methodology developed is robust and justifiable. The numerical method was well established and validated. The results and analysis are thoroughly presented by comparing with previous research results. Personally I enjoyed reading this informative article and learned a lot from it. Below are my review comments:
- What is the size of the mesh element that is fine enough for resolving the smallest scales of the vortices? What is the total number of mesh elements?
- In section 3.1, the authors mentions that "the smallest scales of the vortices that can be fully resolved by the numerical model?", however, in section 4, they say that the "discrepancy between the present experiments and numerical simulations can be traced back to turbulence in the wake of the bubbles that is not captured in the numerical simulations". The authors may want to revise the description of the mesh size in section 3.1.
- To draw the conclusion "This indicates that the effect of the confinement is to reduce the deformation of the bubbles and, correspondingly, increase the mean aspect ratio (line 737 - 738)" one may also need to consider the distance between the wall and the bubble.
- Will the authors consider to study the effect of high viscosity of liquid on the shape and rising velocity of bubbles in a confined channel?
- Line 111, “generated” seems a typo.
- Line 527, "To help interpreting" seems a typo.
- Line 818, "indicate" seems a typo.
